# Rsp5/NEDD4 and ESCRT regulate TDP-43 toxicity and turnover via an endolysosomal clearance mechanism

Aaron Byrd[1,2]* , Lucas J. Marmorale[1]* , Sophia Marcinowski[1] , Megan M. Dykstra[3] , Vanessa Addison[1] , Sami J. Barmada[3] , and J. Ross Buchan[1]

A pathological hallmark in >97% of amyotrophic lateral sclerosis (ALS) cases is the cytoplasmic mislocalization and aggregation of TDP-43, a nuclear RNA-binding protein, in motor neurons. Driving clearance of cytoplasmic TDP-43 reduces toxicity in ALS models, though how TDP-43 clearance is regulated remains controversial. We conducted an unbiased yeast screen using high-throughput dot blotting to identify genes that affect TDP-43 levels. We identified ESCRT complex genes, which induce membrane invagination (particularly at multivesicular bodies; MVBs) and genes linked to K63 ubiquitination (particularly cofactors of the E3 ubiquitin ligase Rsp5; NEDD4 in humans), as drivers of TDP-43 endolysosomal clearance. TDP-43 colocalized and bound Rsp5/NEDD4 and ESCRT proteins, and perturbations to either increased TDP-43 aggregation, stability, and toxicity. NEDD4 also ubiquitinates TDP-43. Lastly, TDP-43 accumulation induces giant MVB-like vesicles, within which TDP-43 accumulates in a NEDD4-dependent manner. Our studies shed light on endolysosomal-mediated cytoplasmic protein clearance, a poorly understood proteostasis mechanism, which may help identify novel ALS therapeutic strategies.

## Introduction

Amyotrophic lateral sclerosis (ALS) is an age-related neurodegenerative disease that affects motor neurons and support cells of the brain and spinal cord. ALS is characterized by progressive degeneration and death of motor neurons, muscle weakness, and fatal paralysis due to respiratory failure that typically occurs 2–5 years after diagnosis. While ~5–10% of cases exhibit a hereditary component, most ALS cases are sporadic and are not associated with a specific gene mutation. Many familial ALS mutations are in genes that impact RNA metabolism and proteostasis, indicating a likely role for these processes in ALS pathology. However, the molecular mechanisms underlying ALS onset and disease progression are still unknown, and no effective treatments for ALS exist (Taylor et al., 2016).

Regardless of disease etiology, a common pathological feature that unites >97% of all ALS cases is the cytoplasmic mislocalization, accumulation, and often aggregation of a nuclear RNA-binding protein, TAR DNA-binding protein 43 (TDP-43) (Ling et al., 2013). TDP-43 preferentially binds UG-rich sequences and functions in many mRNA metabolic processes, including regulation of alternative splicing, mRNA localization, and mRNA stability. TDP-43 mislocalization and accumulation within the cytoplasm results in a toxic gain-of-function, which

could reflect TDP-43 aggregates sequestering key RNAs and proteins, as well as a nuclear loss-of-function, in which TDP-43 nuclear depletion may impair the performance of its various nuclear roles (Cascella et al., 2016).

Failure to effectively clear cytoplasmic TDP-43 may facilitate the pathology of ALS. Indeed, many proteins involved in proteostasis mechanisms such as autophagy and the ubiquitin–proteasome system (UPS), including SQSTM1 (Fecto et al., 2011), VCP (Watts et al., 2004), UBQLN2 (Deng et al., 2011), and OPTN (Maruyama et al., 2010), are mutated in rare familial ALS cases. Also, ALS disease phenotypes can be recapitulated by overexpressing TDP-43 in numerous models (yeast [Johnson et al., 2008], flies [Li et al., 2010], and human cells [Bilican et al., 2012]), indicating that accumulation of TDP-43 is pathogenic. Additionally, increasing TDP-43 degradation by driving bulk, nonselective autophagy can reverse toxicity associated with TDP-43 accumulation and/or mislocalization (Wang et al., 2012; Barmada et al., 2014). Therefore, increasing clearance of TDP-43 may be a broadly applicable therapeutic strategy for diseases with cytoplasmic TDP-43 proteinopathy.

The mechanisms by which TDP-43 is degraded remain highly debated. Prior studies have postulated that insoluble TDP-43

[1]Department of Molecular and Cellular Biology, University of Arizona, Tucson, AZ, USA; [2]Astrazeneca, Gaithersburg, MD, USA; [3]Department of Neurology, University of Michigan, Ann Arbor, MI, USA.

*A. Byrd and L.J. Marmorale contributed equally to this paper. Correspondence to J. Ross Buchan: rbuchan@arizona.edu.



aggregates and oligomers are targeted for clearance via autophagy and soluble TDP-43 is degraded via the UPS (Barmada et al., 2014; Scotter et al., 2014; Wang et al., 2010; Wang et al., 2012; Urushitani et al., 2010). However, these studies often relied on chemical inhibitors of autophagy, which also impair other lysosomal trafficking pathways (e.g., endolysosomal trafficking) or lysosome function generally, making the interpretation of results difficult. Additionally, several E3 ubiquitin ligases can ubiquitinate TDP-43, but the effects of these ubiquitination events are often not correlated to TDP-43 clearance (Hebron et al., 2013; Lee et al., 2018; Hans et al., 2014; Uchida et al., 2016).

We and others have shown in *Saccharomyces cerevisiae* (yeast), human cell culture, and induced pluripotent stem cell (iPSC)-derived motor neurons from ALS patients that cytoplasmic TDP-43 clearance and toxicity are strongly dependent on an autophagy-independent endolysosomal degradation pathway (Hao et al., 2021, *Preprint*; Leibiger et al., 2018; Liu et al., 2017; Liu et al., 2020). Notably, blocks to autophagy and the proteasome showed minimal effects on TDP-43 clearance and toxicity in yeast models. In yeast and human cell models, impairment of endolysosomal flux caused increased TDP-43 accumulation, aggregation, and toxicity, whereas enhancing endolysosomal flux suppressed these phenotypes; fractionation studies in human cells also revealed TDP-43 association with markers of early and late endosomes (Leibiger et al., 2018; Liu et al., 2017; Liu et al., 2020). In a TDP-43 fly model, genetic defects in endocytosis exacerbated motor neuron dysfunction, whereas enhancement of endocytosis suppressed such dysfunction. Lastly, expression of cytoplasmic TDP-43, in particular aggregation-prone mutants, inhibits endocytosis (Liu et al., 2017; Liu et al., 2020).

Given the lack of consensus on how TDP-43 is cleared and which clearance pathways are most relevant to ALS phenotypes, we performed an unbiased genetic screen in yeast to determine genes that alter TDP-43 abundance. We discovered conserved proteins that regulate TDP-43 toxicity, aggregation, and protein stability by targeting TDP-43 for endolysosomal clearance, specifically endosomal sorting complexes required for transport (ESCRT) proteins, and the E3 ubiquitin ligase Rsp5 (yeast)/NEDD4 (human). NEDD4 also binds, ubiquitinates, and destabilizes TDP-43 in several cell types, including iPSC-derived neurons (iNeurons). Finally, mild TDP-43 overexpression (twofold, including endogenous TDP-43) causes accumulation of giant Rab7/CD63-positive vesicles, within which TDP-43 accumulates in a NEDD4-driven manner. Collectively, these data and prior work suggest that cytoplasmic TDP-43 is cleared by an endolysosomal mechanism and that pathological accumulation of TDP-43 may disrupt endolysosomal trafficking.

## Results

### A dot-blot genetic screen in yeast identifies genes regulating TDP-43 steady-state levels

To our knowledge, an unbiased genetic screen to identify regulators of TDP-43 abundance has not been conducted in any model system. Such a screen could identify regulators of both TDP-43 synthesis and degradation and, in turn, novel therapeutic targets for diseases characterized by TDP-43 proteinopathy. Therefore, we designed a dot-blot genetic screen, a refinement of a previously published method (Zacchi et al., 2017) that allowed us to measure steady-state TDP-43 protein levels in each strain of the yeast nonessential gene deletion library (Winzeler et al., 1999) (Fig. 1 A). To account for variations in expression due to intrinsic or extrinsic noise, four control strains were used in two different locations on each 384-well dot-blot plate. First, a WT strain containing a single copy TDP-43-YFP plasmid was included as a baseline reference. Second, a bre5Δ strain containing the same TDP-43-YFP plasmid ensured that strains with lower TDP-43 protein levels were detectable; yeast lacking Bre5, a deubiquitinase cofactor, exhibit significantly reduced steady-state TDP-43 protein levels (Liu et al., 2020). Third, a WT strain containing a high-copy TDP-43-GFP plasmid ensured strains with higher TDP-43 protein levels were detectable. Lastly, a WT strain containing an empty vector controlled for nonspecific binding of the α-GFP/YFP antibody (Fig. 1 B). Besides α-GFP/YFP labeling, blots were also probed with an α-GAPDH antibody, which mitigated variable detection of TDP-43 caused by differences in strain growth rates or lysis efficiencies (Fig. 1 B).

Altogether, this screen generated 196 hits total, with 44 strains exhibiting significantly increased TDP-43 abundance (>1.5-fold increase, P value <0.05) and 152 strains exhibiting significantly decreased TDP-43 abundance (<0.5-fold decrease, P value <0.05; Table S1). Encouragingly, our screen reidentified previously known regulators of TDP-43 abundance in yeast, such as bre5Δ (30% of WT) (Liu et al., 2020). Other expected regulators of TDP-43 abundance obtained in the screen included regulators of carbon metabolism, consistent with the *GAL1*-driven nature of the TDP-43 construct used. Furthermore, two separate rim8Δ strains, present twice in the deletion library owing to a gene annotation error, were both identified as hits with very similar levels of TDP-43 accumulation (5.0- and 5.5-fold accumulation).

Overall, gene ontology analysis indicated that terms most significantly enriched among the hits from the screen included K63-linked ubiquitination, vacuolar proton-transporting V-type ATPase complex assembly, and protein localization to endosomes (Fig. 1 C). Among the top 10 hits with increased TDP-43 abundance were rim8Δ, mms2Δ, vps28Δ, ygr122wΔ, and rad6Δ strains, which we validated manually via western blotting (Fig. 1 D). Rim8 is an adaptor in the endolysosomal pathway for the E3 ubiquitin ligase Rsp5, which facilitates targeting of Rsp5 to its substrates for ubiquitination (Fang et al., 2014). Rim8 also binds the ESCRT-I complex to target its substrates (typically transmembrane proteins) for eventual degradation in the vacuole (Herrador et al., 2010). Mms2 and Rad6 are E2 ubiquitin-conjugating enzymes that both function in K63-linked ubiquitination (Wen et al., 2012; Simões et al., 2022). Vps28 is a core component of the ESCRT-I complex involved in sorting cargoes into MVBs (Katzmann et al., 2001). Ygr122w is a protein of unclear function that binds the ESCRT-III complex factor Snf7 and genetically interacts with Rsp5 (Rothfels et al., 2005). Together, combined with our prior work (Liu et al., 2017; Liu et al., 2020),

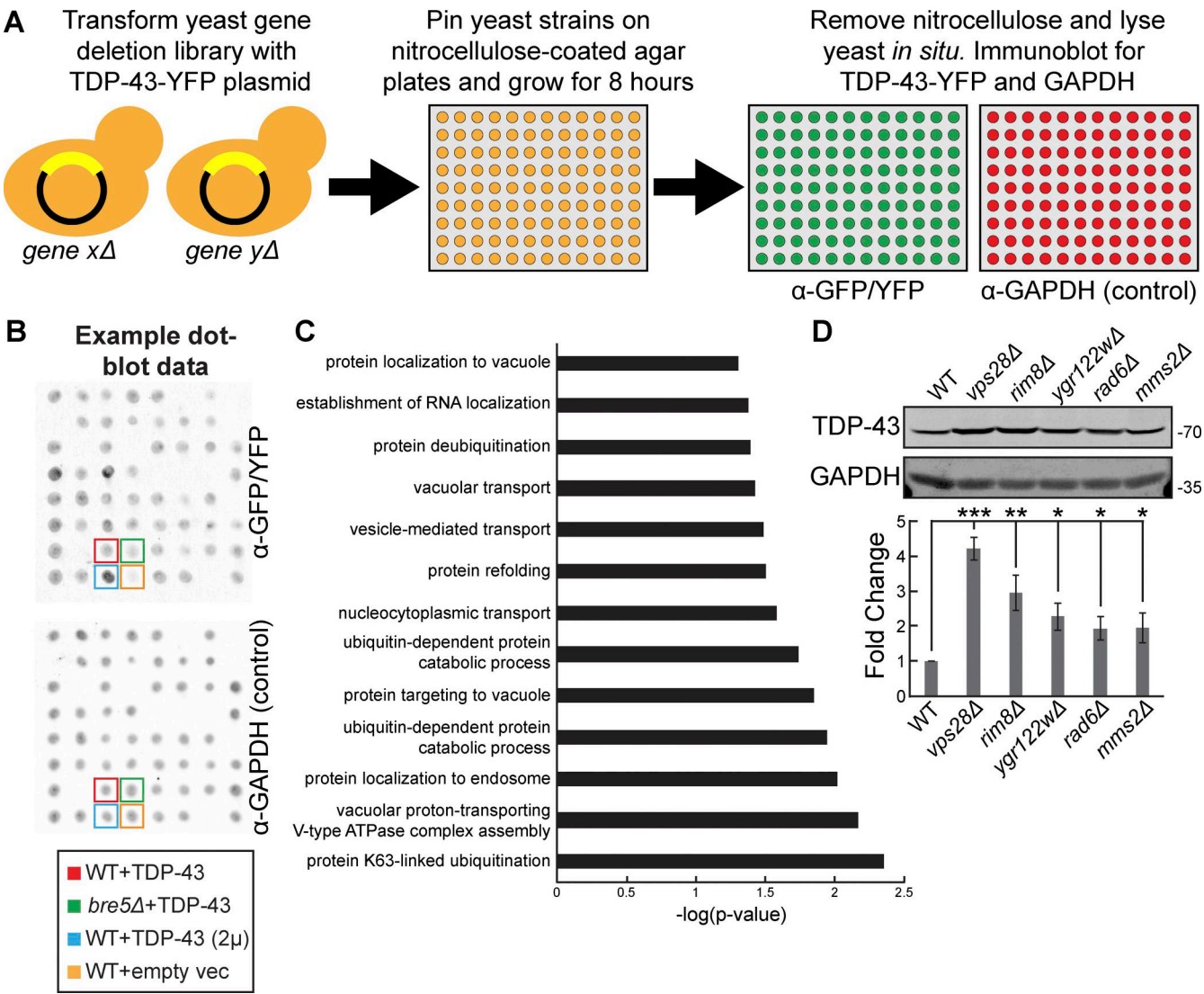

Figure 1. **A dot-blot genetic screen in yeast identifies genes regulating TDP-43 steady-state levels. (A)** Schematic representation of the dot-blot genetic screen. **(B)** Example dot-blot with control strains and placement indicated. **(C)** Gene ontology terms of biological functions significantly enriched within the hits from the dot-blot genetic screen. **(D)** Indicated strains were transformed with TDP-43-mRuby2 plasmid (pRB194) and grown to mid-logarithmic growth phase, and TDP-43-mRuby2 was detected by western blot using anti–TDP-43 antibody. TDP-43 levels were normalized to GAPDH loading control. *, P < 0.05; **, P < 0.01; ***, P < 0.001 by one-way analysis of variance with Tukey's post hoc test; N = 3. Error bars indicate standard error of the mean. Source data are available for this figure: SourceData F1.

this led us to a working model that TDP-43 is subject to K63-linked ubiquitination via Rsp5, which then ultimately targets it to the ESCRT complex for internalization into MVBs.

**TDP-43 clearance and aggregation depend on ESCRT function**

Previously, we showed that mutations in ESCRT genes increase TDP-43 toxicity (Liu et al., 2017). To verify the defects in TDP-43 clearance in ESCRT mutants as suggested by our screen, we measured protein levels of TDP-43 in mutants of each ESCRT subcomplex via western blotting and found that almost all these mutant strains had significantly increased steady-state levels of TDP-43 versus a WT strain (Fig. 2 A). We then determined if these increased levels of total TDP-43 protein correlated with an increase in TDP-43 aggregation via fluorescence microscopy. Indeed, relative to WT, strains that exhibited small increases in

TDP-43 abundance also had small increases in TDP-43 foci per cell (vps4Δ and vps2Δ), while strains that had large increases in TDP-43 abundance had significantly increased TDP-43 foci per cell (vps28Δ, vps36Δ, and vps27Δ; Fig. 2 B). To determine whether increases in TDP-43 abundance were due to improper clearance, rather than altered synthesis, we performed cycloheximide chase assays where global protein synthesis was halted by the cycloheximide addition, followed by whole yeast lysate collection at different time points over 24 h, to determine how quickly TDP-43 is cleared. TDP-43 stability was significantly increased in the ESCRT mutant strains vps36Δ and vps4Δ (Fig. 2 C). TDP-43 foci also colocalized with the core ESCRT factors Vps28, Vps36, and Vps4 (Fig. 2 D); whether these represent functional TDP-43–ESCRT interactions in clearance or instances of TDP-43 aggregates sequestering ESCRT proteins remains unclear. In

Figure 2. **TDP-43 clearance and aggregation depend on ESCRT function. (A)** Indicated strains were transformed with TDP-43-mRuby2 plasmid (pRB194) and grown to mid-logarithmic growth phase, and TDP-43-mRuby2 was detected by western blot using anti–TDP-43 antibody. TDP-43 levels were normalized to GAPDH loading control. **, P < 0.01; ***, P < 0.001 by one-way analysis of variance with Tukey's post hoc test; N = 3. **(B)** Indicated strains were transformed with TDP-43-mRuby2 plasmid and imaged in mid-logarithmic growth phase with foci/cell quantification below. ***, P < 0.001 by one-way analysis of variance with Tukey's post hoc test; N = 3. Scale bar, 5 μm. **(C)** Indicated strains were transformed with TDP-43-mRuby2 plasmid and grown to mid-logarithmic growth phase before being treated with 0.2 mg/ml cycloheximide for 0, 6, 12, and 24 h. TDP-43-mRuby2 levels were detected by western blot using anti–TDP-43 antibody and normalized to GAPDH loading control. **, P < 0.01; ***, P < 0.001 by two-way analysis of variance with Tukey's post hoc test; N = 3. **(D)** Strains endogenously expressing GFP-Vps28, GFP-Vps36, and GFP-Vps4 were transformed with a TDP-43-mRuby2 plasmid and imaged at mid-log phase growth. Percentage values indicate colocalization frequency of TDP-43-mRuby2 foci with GFP foci. N = 3; scale bar, 5 μm. Error bars in all graphical panels represent standard error of the mean. Source data are available for this figure: SourceData F2.

summary, these data strongly suggest that TDP-43 clearance in yeast depends upon the ESCRT complex.

### TDP-43 toxicity, aggregation, and clearance depend on the E3 ubiquitin ligase Rsp5

Since factors related to K63 ubiquitination and known Rsp5 interactors were among the top hits of our dot-blot screen, we investigated the role of the E3 ubiquitin ligase Rsp5 in the toxicity, aggregation, and clearance of TDP-43. Rsp5 is one of the major endolysosomal E3 ubiquitin ligases in yeast responsible

for K63-linked ubiquitination and has many functions in regulating protein turnover and trafficking (Sardana and Emr, 2021). *RSP5* is an essential gene; thus, we used a temperature-sensitive mutant, *rsp5-3*, which possesses three missense point mutations, T104A, E673G, and Q716P, that inhibit Rsp5's catalytic activity at elevated temperatures (Neumann et al., 2003). TDP-43 toxicity, evidenced via serial dilution spotting assays, was greatly exacerbated by partial inactivation of *rsp5-3* by temperature shift to 30°C (Fig. 3 A). Even at the "permissive" temperature of 25°C, the *rsp5-3* strain exhibited greater growth defects from TDP-43

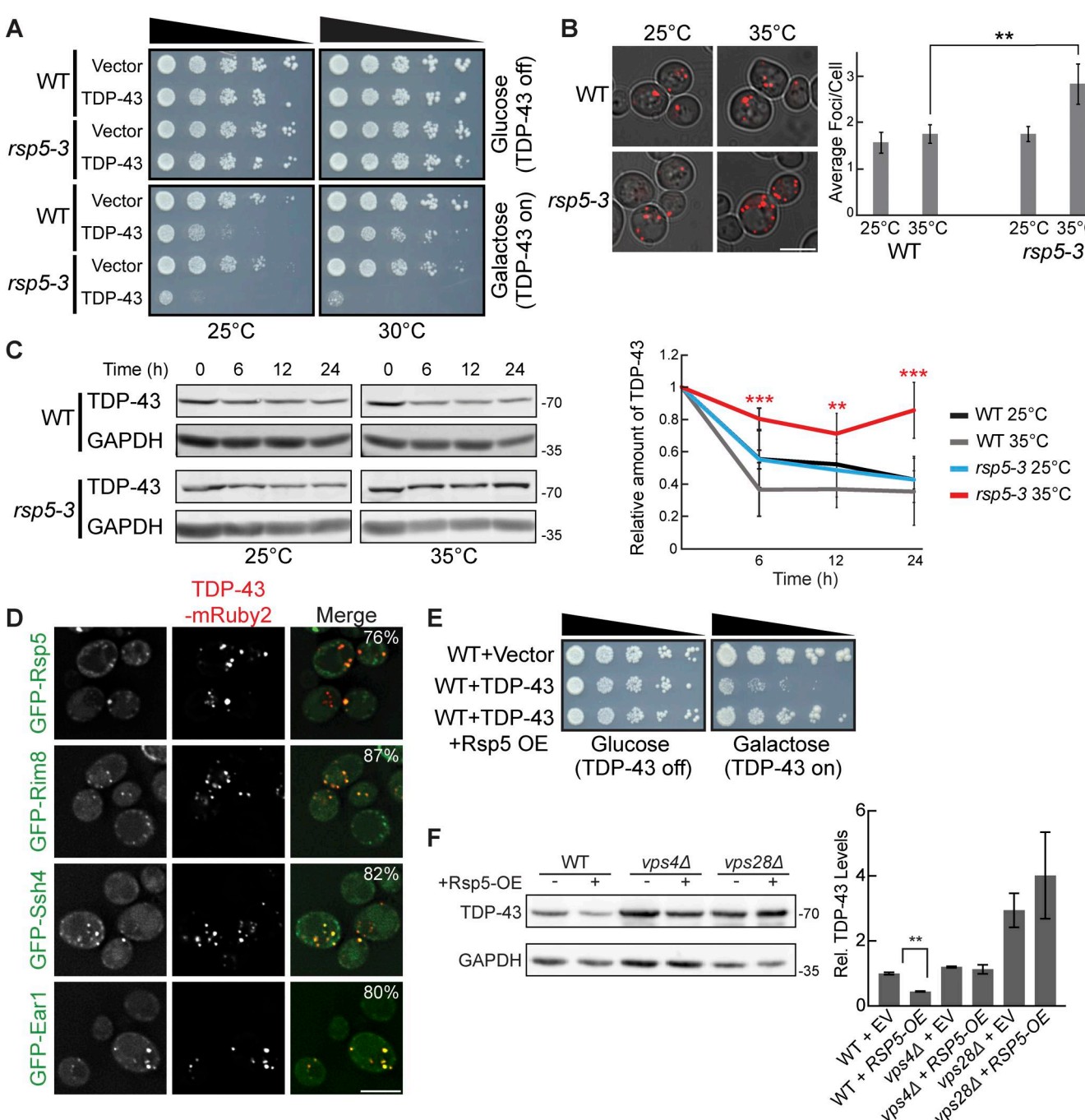

Figure 3. **TDP-43 toxicity, aggregation, and toxicity depend on the E3 ubiquitin ligase Rsp5. (A)** Representative serial dilution growth assay of WT and *rsp5-3* strains transformed with the empty vector (pRB43) or untagged TDP-43 plasmid (pRB232) grown at 25°C or 30°C as indicated. *N* = 3. **(B)** WT and *rsp5-3* strains were transformed with TDP-43-mRuby2 plasmid and grown to mid-logarithmic growth phase at 25°C, and samples were imaged. Cultures were then incubated at 35°C for 16 h and imaged again. **, P < 0.01 by one way analysis of variance with Tukey's post hoc test; *N* = 3. Scale bar, 5 μm. **(C)** WT and *rsp5-3* strains were transformed with TDP-43-mRuby2 plasmid and grown to mid-logarithmic growth phase at 25°C before being treated with 0.2 mg/ml cyclo-heximide for 0, 6, 12, and 24 h. At time 0, cultures were either shifted to 35°C or kept at 25°C for the remainder of the time course. TDP-43-mRuby2 levels were detected by western blot using anti–TDP-43 antibody and normalized to GAPDH loading control. **, P < 0.01; ***, P < 0.001 by one-way analysis of variance with Tukey's post hoc test; *N* = 3. **(D)** Strains endogenously expressing GFP-Rsp5, GFP-Rim8, GFP-Ssh4, and GFP-Ear1 were transformed with a TDP-43-mRuby2 plasmid (pRB194) and imaged at mid-log phase growth. Percentage values indicate colocalization frequency of TDP-43-mRuby2 foci with GFP foci. *N* = 3; scale bar, 5 μm. **(E and F)** Representative serial dilution growth assay of WT strain transformed with an empty vector control (pRB43) or untagged TDP-43 plasmid (pRB232) and with the empty vector control (pRB511) or Rsp5 overexpression plasmid (pRB476). *N* = 3 (F) WT, *vps4Δ*, and *vps28Δ* strains were transformed with TDP-43-YFP (pRB109) and also transformed with Rsp5 overexpression (pRB476) or empty vector control plasmid (pRB511) and grown to mid-logarithmic growth phase. TDP-43-YFP (pRB109) was detected by western blot using anti–TDP-43 antibody. TDP-43 levels were normalized to GAPDH loading control. **, P < 0.01 by Welch's two-tailed T test; *N* = 3. Error bars in all graphical panels represent standard error of the mean. Source data are available for this figure: SourceData F3.

expression, which may reflect partially defective Rsp5 function. Next, we examined if enhanced TDP-43 toxicity in the *rsp5-3* strain correlated with an increase in TDP-43 accumulation and aggregation. As expected, we saw an increase in TDP-43 aggregation when *rsp5-3* was strongly inactivated by shifting the temperature to 35°C, while comparable amounts were seen at the permissive temperature of 25°C (Fig. 3 B). We next determined whether Rsp5 function is important for TDP-43 clearance by performing cycloheximide chase assays. In the WT and *rsp5-3* strains grown at 25°C, TDP-43 abundance decreased similarly over a 24-h time course. However, when strains were shifted to 35°C, the *rsp5-3* strain exhibited significantly greater TDP-43 stability than the WT strain (Fig. 3 C). Additionally, Rsp5 co-localizes in TDP-43 cytoplasmic foci (Fig. 3 D). Notably, when we overexpressed Rsp5 in a WT strain, TDP-43 toxicity and steady-state protein levels were significantly reduced (Fig. 3, E and F). This reduction was attenuated when Rsp5 was overexpressed in *vps28Δ* and *vps4Δ* strains, indicating that Rsp5 relies on ESCRT function to promote TDP-43 turnover.

Rsp5 substrate specificity often requires adaptor proteins of the arrestin-related trafficking adaptor family (Sardana and Emr, 2021). Among the best-known Rsp5 adaptors are Rim8, Ear1, and Ssh4, which are involved in sorting Rsp5 substrates to the endolysosomal pathway (Sardana and Emr, 2021). Rim8 was a top repeated hit in our dot-blot screen, having a > fivefold increase in TDP-43 abundance (Table S1). Ear1 and Ssh4 often function redundantly, but both are involved in cargo sorting of proteins delivered to MVBs and were therefore of interest to our study. TDP-43 colocalizes with all three of these Rsp5 adaptor proteins via fluorescence microscopy (Fig. 3 D). To address the importance of all Rsp5 adaptors in the regulation of TDP-43 levels, we performed steady-state western blotting of TDP-43 abundance in all 21 Rsp5 adaptor nulls. Only *rim8Δ* showed a significant effect on TDP-43 clearance (Fig. S1). In summary, Rsp5 regulates TDP-43 turnover and toxicity in our yeast model, likely aided in part by Rim8.

## Dominant-negative VPS4A^E228Q increases endogenous TDP-43 stability, cytoplasmic mislocalization, and aggregation in human cells

To examine the role of ESCRT-dependent turnover of TDP-43 in human cells, we used HEK293A cells and a dominant-negative mutant of the core ESCRT protein, VPS4A. This mutant possesses a point mutation, E228Q, that prevents ATP hydrolysis and greatly hinders ESCRT functions (Votteler et al., 2016). We first determined whether EGFP-VPS4A^E228Q expression affected the turnover of endogenous TDP-43 by performing cycloheximide chase assays. We found that endogenous TDP-43 in cells transfected with EGFP-VPS4A^E228Q was significantly more stable than in untreated WT cells (Fig. 4 A). Since TDP-43 colocalized with Vps4 in yeast, we investigated whether TDP-43 colocalized with EGFP-VPS4A^E228Q by performing immunofluorescence of endogenous TDP-43. We often observed colocalization of TDP-43 with EGFP-VPS4A^E228Q in cytoplasmic foci, some of which took on vesicular-like patterns of colocalization, though we also observed instances of TDP-43 and EGFP-VPS4A^E228Q in separate foci/vesicular structures (Fig. 4 B). Endogenous TDP-43 levels in

both nuclei and the cytoplasm also increased significantly in the VPS4A^E228Q background (Fig. 4, B and C). TDP-43 was significantly more likely to form cytoplasmic foci in cells transfected with EGFP-VPS4A^E228Q versus WT cells (Fig. 4 D). Lastly, decreased cell proliferation associated with overexpression of TDP-43-GFP and/or the C-terminal fragment (CTF) and disease-associated isoform, TDP-35-GFP, was significantly exacerbated by expression of VPS4A^E228Q (Fig. 4 E). Collectively, these data suggest that the ESCRT complex facilitates TDP-43 turnover and, in turn, regulates TDP-43 toxicity.

## NEDD4 physically interacts with and regulates TDP-43 stability and toxicity

We next determined if NEDD4, the Rsp5 homolog in humans, affects clearance of TDP-43 in a mild overexpression context. We used HEK293A cells expressing stably integrated TDP-43-GFP at comparable levels with endogenous TDP-43 (which is also present) in HEK293A cells; this also results in a slight increase in cytoplasmic TDP-43 localization (Liu et al., 2017). By performing a cycloheximide chase assay, we found that TDP-43-GFP stability was significantly increased in cells where NEDD4 expression was knocked down ~50–60% versus WT (Fig. 5 A and Table S2). Additionally, overexpression of NEDD4 via NEDD4-mCherry plasmid transfection significantly lowered the steady-state levels of TDP-43-GFP (Fig. 5 B). These data indicate that TDP-43 stability and abundance are regulated by NEDD4.

TDP-43 cytoplasmic abundance tends to correlate with induced cellular toxicity; therefore, we investigated whether altering TDP-43 levels and stability by overexpression or knockdown of NEDD4 affected cell proliferation. Indeed, decreased cell proliferation associated with overexpression of TDP-43-GFP and TDP-35-GFP was significantly rescued by overexpression of NEDD4 (Fig. 5 C). When NEDD4 was knocked down, TDP-43-GFP and TDP-35-GFP expression significantly reduced cell proliferation (Fig. 5 C). Thus, NEDD4 is a negative regulator of TDP-43 toxicity.

To determine whether the effects of NEDD4 overexpression involved interaction with TDP-43, we performed live-cell fluorescence microscopy of cells overexpressing NEDD4-mCherry and TDP-43-GFP. We found that NEDD4-mCherry and TDP-43-GFP colocalize in cytoplasmic foci (Fig. 5 D). *In vivo* binding of NEDD4-mCherry and TDP-43-GFP was also detected via immunoprecipitation (Fig. 5 E).

These data indicated that NEDD4 interacts with and facilitates turnover of TDP-43-GFP in HEK293A cells, thus likely explaining NEDD4's mitigating effects on TDP-43-GFP toxicity.

## NEDD4 polyubiquitinates TDP-43, in part via K63-linked ubiquitin chains

Since NEDD4 is an E3 ubiquitin ligase, and ubiquitination is commonly involved in protein degradation, we performed a ubiquitination assessment of immunopurified TDP-43-GFP from cells with NEDD4 overexpression or NEDD4 knockdown. In NEDD4 overexpression cells, we observed a consistent increase in the polyubiquitination of TDP-43-GFP versus control (Fig. 5 F). Conversely, in NEDD4 siRNA knockdown cells, TDP-43-GFP ubiquitination was markedly decreased (Fig. 5 F).

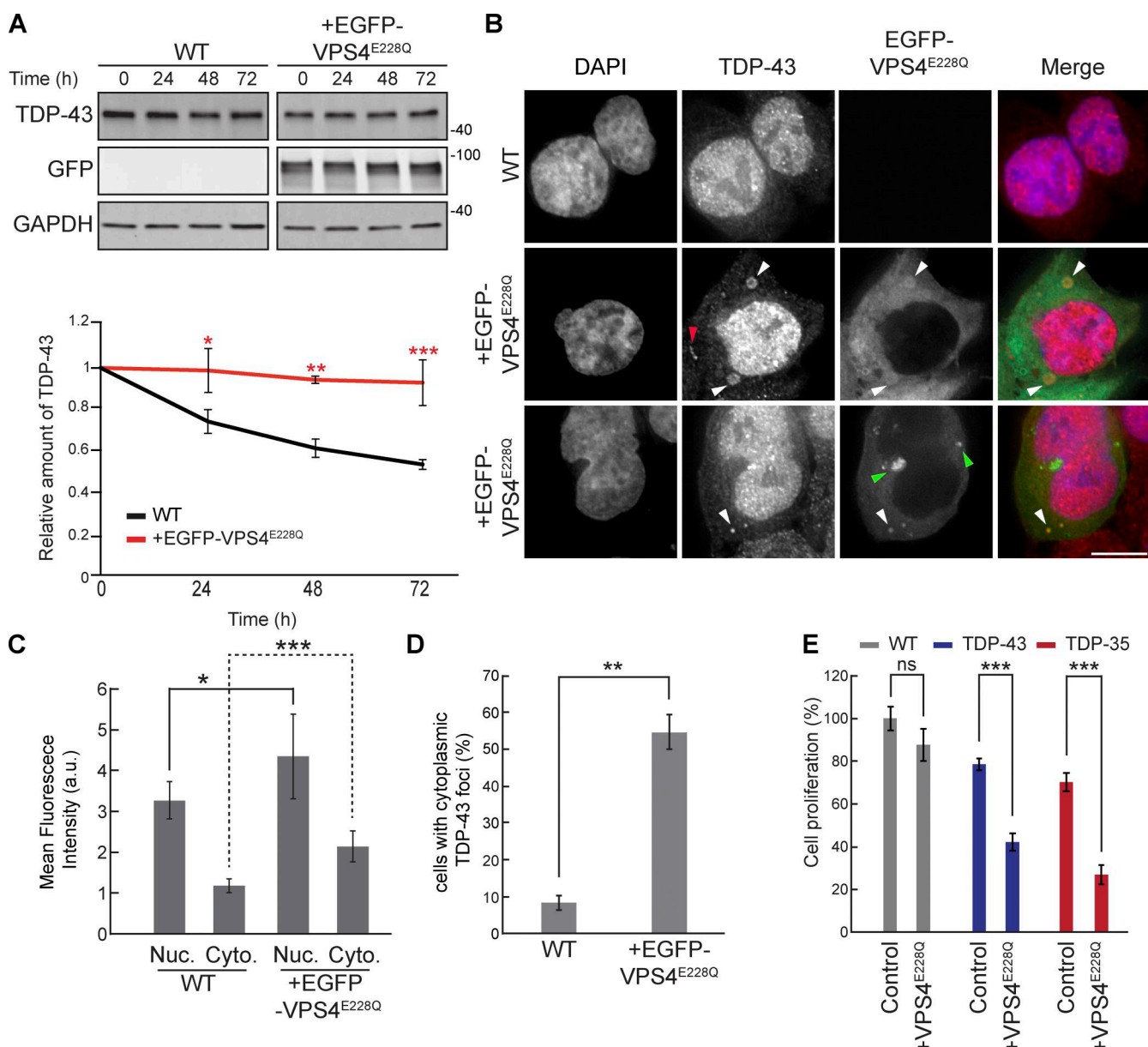

Figure 4. **VPS4A^{E228Q} increases TDP-43 stability and promotes cytoplasmic mislocalization. (A)** HEK293A WT cells were transfected with EGFP-VPS4A^{E228Q} plasmid (pRB475) or mock transfected and grown in normal conditions for 48 h. Cells were then treated with 0.3 mg/ml cycloheximide for the indicated amounts of time, and TDP-43 levels were detected by western blot with anti–TDP-43 antibodies and normalized to the level of the GAPDH loading control. *, P < 0.05; **, P < 0.01; ***, P < 0.001 by one-way analysis of variance with Tukey's post hoc test; N = 3. **(B)** HEK293A WT cells were either mock transfected (top row) or transfected with EGFP-VPS4A^{E228Q} plasmid (next two rows, to show phenotypic diversity) and grown in normal conditions for 48 h before being fixed and immunostained with anti–TDP-43 antibodies. White arrowheads indicate TDP-43 and EGFP-VPS4A^{E288Q} colocalization; red arrowhead indicates example of TDP-43-distinct foci; green arrowheads indicate examples of distinct EGFP-VPS4A^{E288Q} foci/vesicular structures. N = 3; scale bar, 10 µm. **(C)** Quantification of mean nuclear and cytoplasmic TDP-43 signal intensities in HEK293A WT cells and cells transfected with EGFP-VPS4A^{E228Q} plasmid. *, P < 0.05; ***, P < 0.001 by Mann–Whitney U test (nuclear) or Welch's two-tailed T test (cytoplasmic); N = 30. **(D)** Quantification of the percentage of cells with cytoplasmic TDP-43 foci in HEK293A WT cells and cells transfected with EGFP-VPS4A^{E228Q} plasmid. **, P < 0.01 by Welch's two-tailed T test; N = 3. **(E)** HEK293A, HEK-TDP-43-GFP, and HEK-TDP-35-GFP cells were seeded at equal cell numbers and transfected with VPS4A^{E228Q} plasmid or mock transfected, and cell proliferation was compared after 48 h ***, P < 0.001 by one-way analysis of variance with Tukey's post hoc test; N = 3. Error bars in all graphical panels represent standard error of the mean. Source data are available for this figure: SourceData F4.

Like Rsp5, NEDD4 has primarily been associated with K63-linked ubiquitination ([Maspero et al., 2013](); [Tofaris et al., 2011]()), though an ability to add K48-linked ubiquitin chains to substrates (more commonly associated with proteasomal degradation) has also been reported ([Lohraseb et al., 2022](); [Huang et al.,

[2019]()). We were also curious if our observed NEDD4 effects on TDP-43 ubiquitination might be specific to overexpressed or GFP-tagged TDP-43. Thus, we also examined immunopurified endogenous TDP-43 in WT or NEDD4 overexpressing cells for evidence of alterations in polyubiquitination (using a pan-Ub

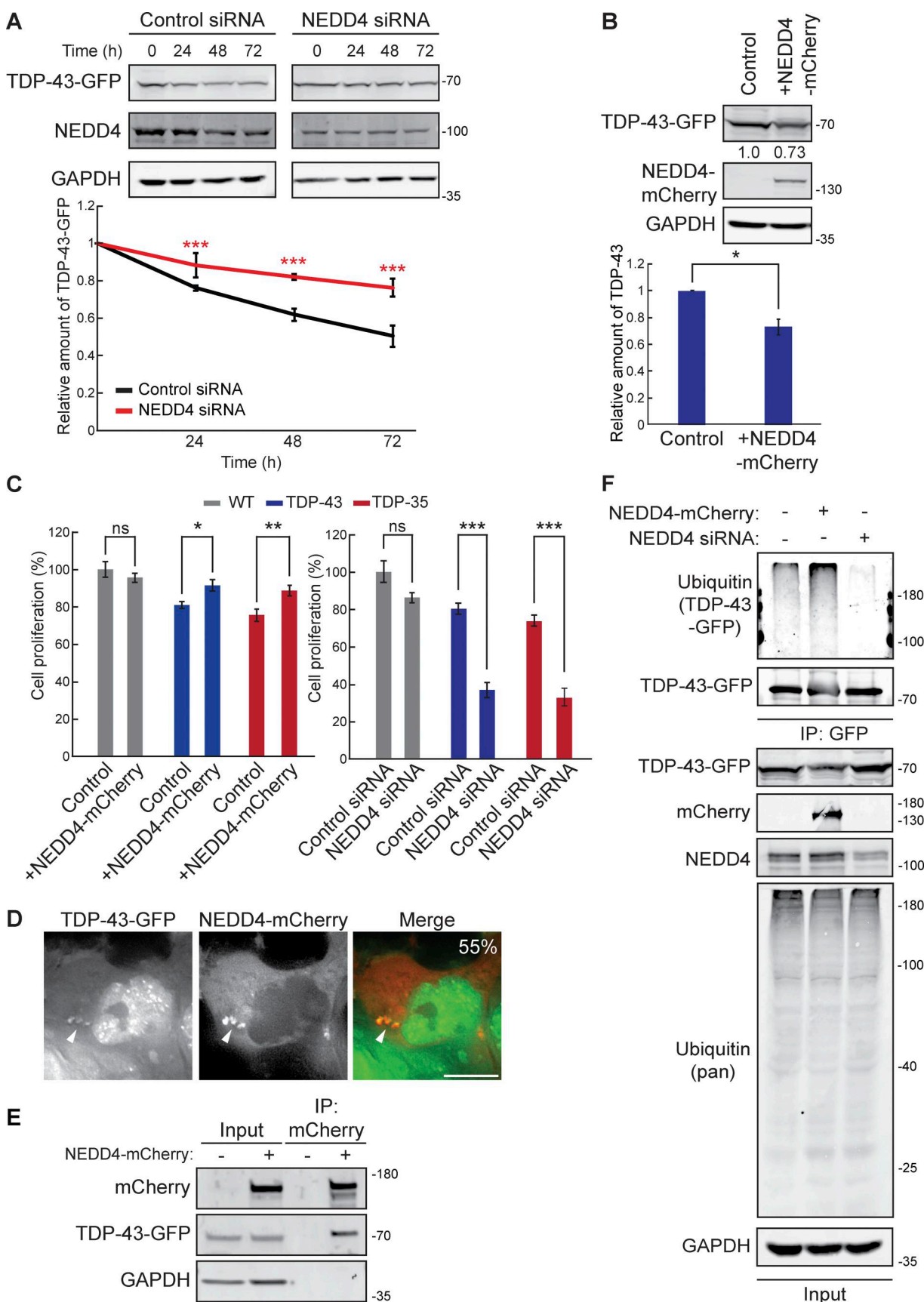

Figure 5.  **NEDD4 regulates TDP-43 turnover, toxicity, and ubiquitination. (A)** HEK-TDP-43-GFP cells were transfected with siRNAs targeting NEDD4 or noncoding control and grown in normal conditions for 3 days. Cells were then treated with 0.3 mg/ml cycloheximide for the indicated amounts of time, and

TDP-43-GFP levels were detected by western blot with anti-GFP antibodies and normalized to the level of the GAPDH loading control. ***, P < 0.001 by two-way analysis of variance with Tukey's post hoc test; N = 3. **(B)** HEK-TDP-43-GFP cells were transfected with NEDD4-mCherry plasmid (pRB334) and grown in normal conditions for 3 days. TDP-43-GFP levels were detected by western blot with anti-GFP antibodies and normalized to the level of the GAPDH loading control. *, P < 0.05 by Welch's two-tailed T test; N = 3. **(C)** HEK293A, HEK-TDP-43-GFP, and HEK-TDP-35-GFP cells were seeded at equal cell numbers and transfected with NEDD4-mCherry, a mock transfection control (left), or siRNA targeting NEDD4 or noncoding control (right), and cell proliferation was compared after 48 h *, P < 0.05; **, P < 0.01; ***, P < 0.001 by one-way analysis of variance with Tukey's post hoc test; N = 3. **(D)** HEK-TDP-43-GFP cells were transfected with NEDD4-mCherry plasmid and incubated for 24 h before being imaged. Percentage value indicates colocalization frequency of cytoplasmic TDP-43-GFP foci with NEDD4-mCherry foci. N = 3; scale bar, 10 μm. **(E)** HEK-TDP-43-GFP cells were transfected with NEDD4-mCherry plasmid or a mock transfection and incubated for 48 h before mCherry was immunoprecipitated, and TDP-43-GFP binding was assessed via western blot using anti–TDP-43 antibodies. **(F)** HEK-TDP-43-GFP cells were transfected with siRNAs targeting NEDD4, NEDD4-mCherry plasmid, or a mock control and grown for 3 days before TDP-43-GFP was immunoprecipitated under denaturing conditions (See Materials and methods). Error bars in all graphical panels represent standard error of the mean. Source data are available for this figure: SourceData F5.

Ab) and K63- or K48-linked ubiquitination using linkage-specific antibodies. Consistently, we observed an increase in K63-polyubiquitination of TDP-43 driven by NEDD4 overexpression, whereas K48 polyubiquitination of TDP-43 signal, which was challenging to detect, decreased modestly following NEDD4 overexpression (Fig. S2 A). Mimicking our prior data with immunopurified TDP-43-GFP (Fig. 5 F), NEDD4-dependent increases in polyubiquitination were also seen with immunopurified endogenous TDP-43 (Fig. S2 B).

These data indicate that NEDD4 polyubiquitinates TDP-43, at least in part by adding K63-linked ubiquitin chains.

### NEDD4 knockdown inhibits TDP-43 interaction with VPS4A

To further investigate the role of NEDD4 in TDP-43 turnover via an ESCRT-dependent mechanism, without the caveat of possible overexpression artefacts of factors of interest, we immunopurified endogenous TDP-43 and assessed its interaction with endogenous VPS4A. This was conducted in the presence or absence of NEDD4 siRNA knockdown, and the lysosomotropic agent chloroquine (CHQ), which increases pH within endolysosomal compartments, thus impairing endolysosomal degradation. Our rationale was that CHQ might increase detection of otherwise transient TDP-43–VPS4A interactions during the degradation process.

Notably, TDP-43 interacted with VPS4A, an interaction that significantly increased in the presence of CHQ (Fig. S3). TDP-43 interaction with VPS4A was significantly reduced by NEDD4 knockdown (in the absence of CHQ). This suggests that NEDD4, likely via TDP-43 ubiquitination, facilitates TDP-43 interaction with ESCRT factors, including VPS4A.

Finally, our prior demonstration of TDP-43 interaction with NEDD4 involved overexpression of both factors (Fig. 5 E). With endogenous levels of both TDP-43 and NEDD4, we detected a very weak interaction, which increased significantly following CHQ treatment (Fig. S3).

In summary, endogenous TDP-43 interacts with both NEDD4 and VPS4A, particularly when endolysosomal degradation is impaired. NEDD4 also facilitates TDP-43 interaction with VPS4A.

### NEDD4 knockdown increases the stability of endogenous TDP-43 in human neurons

To assess the relevance of NEDD4 activity to TDP-43 stability in a more physiologically relevant ALS model, we used human

iNeurons that have been genetically modified such that endogenous TDP-43 is labeled with the photoconvertible protein Dendra2. Using an optical pulse labeling approach (Fig. 6 A), TDP-43-Dendra2 was photoswitched irreversibly from green to red upon exposure to blue light; subsequent disappearance of red TDP-43-Dendra2 signal over time allowed calculation of TDP-43 half-life. Using this cell line, we knocked down NEDD4 ~60% (Fig. 6 B) using virally transduced shRNA to assess the role of NEDD4 on TDP-43-Dendra2 half-life. NEDD4 knockdown significant increased endogenous TDP-43-Dendra2 half-life from 47.2 to 63.5 h (Fig. 6, C and D). These data suggest NEDD4 facilitates TDP-43 turnover in neurons.

### TDP-43 accumulation causes appearance of enlarged Rab7/CD63-positive vesicular organelles

Since TDP-43 clearance via an ESCRT-driven process suggested, together with our prior data (Liu et al., 2017; Liu et al., 2020), that TDP-43 may be targeted to MVBs, we investigated this possibility via immunofluorescence microscopy using the common MVB marker, Rab7. Surprisingly, TDP-43-GFP and TDP-35-GFP cell lines (which modestly overexpress TDP-43 [Liu et al., 2017]) developed giant vesicles with intense Rab7 signal (Fig. 7 A). Whereas typically MVBs reach about 0.5–1 μm in diameter (Bartheld and Altick, 2011), we commonly observed Rab7 vesicles much larger in diameter (>2.5 μm). These giant Rab7-positive vesicles were significantly enriched in TDP-43-GFP and TDP-35-GFP cells versus WT HEK293A cells (Fig. 7 B), averaging 4.5–5 μm in diameter (Fig. 7 C). Rab7-positive vesicles in our TDP-43-GFP–expressing cell lines were highly variable in size, with some >10 μm in diameter (Fig. 7 D).

To further investigate the compositional nature of the giant Rab7 vesicles in TDP-43-GFP and TDP-35-GFP cells, we performed co-immunostaining of Rab7 along with LC3B, Rab5, or LAMP1 to determine if these putative giant MVBs were possibly hybrid compartments with autophagosomes, early endosomes, or lysosomes, respectively (Fig. 7, D and E). None of these markers strongly colocalized with the Rab7 membrane signal, although we sometimes observed very weak colocalization of each marker (most commonly LC3B) with small sections of the Rab7-positive membrane (Fig. 7, D and E), suggesting a possible amphisome-like quality (Fader and Colombo, 2009; Ganesan and Cai, 2021). We also performed co-immunostaining of Rab7 with CD63, both in WT and TDP-43-GFP–expressing cells. CD63 is a transmembrane protein of the tetraspanin family, which labels

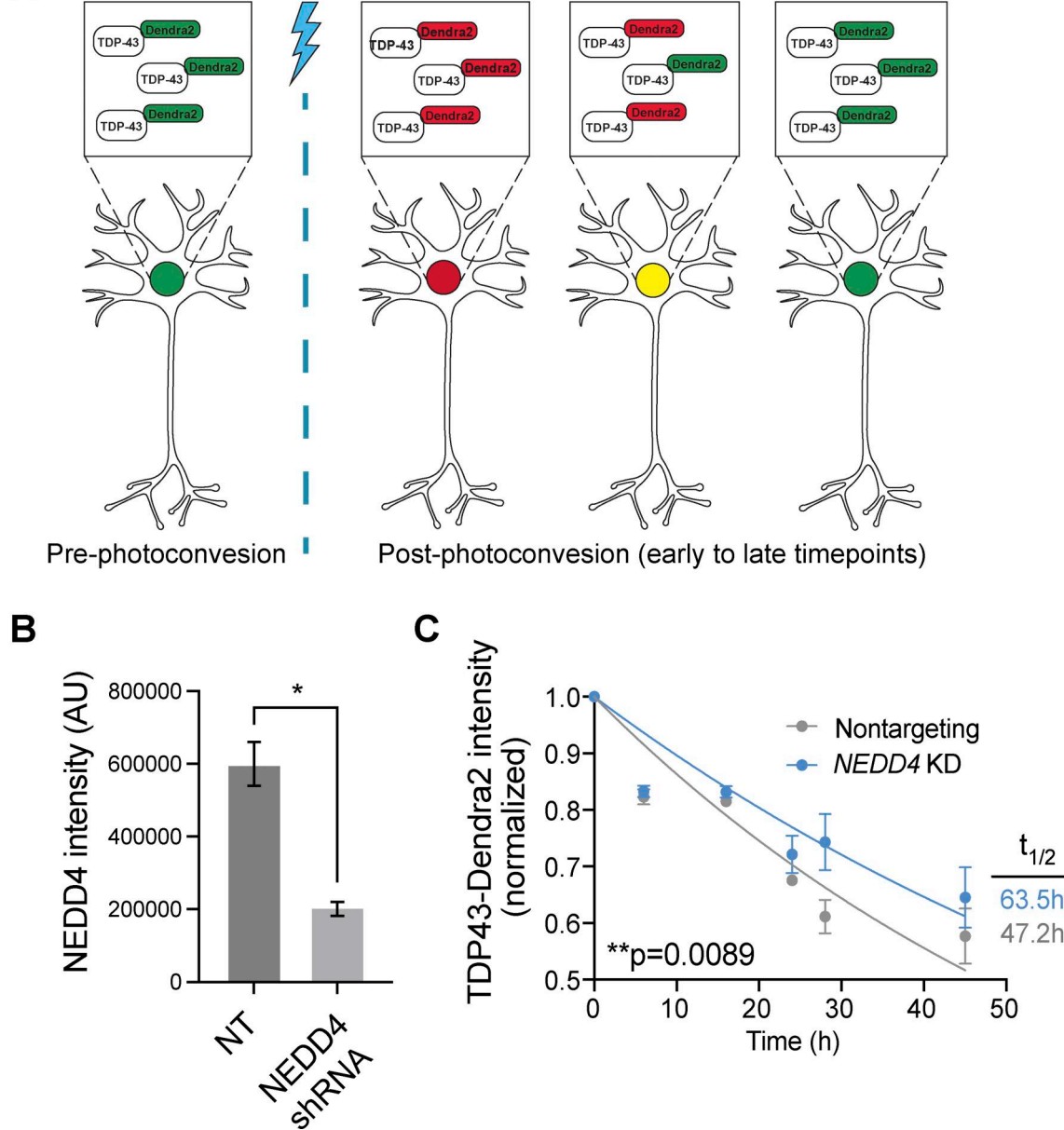

Figure 6. **NEDD4 knockdown stabilizes TDP-43 in iNeurons. (A)** Schematic of optical pulse labeling strategy for measuring TDP-43-Dendra2 half-life. **(B)** NEDD4 knockdown efficiency (lentiviral shRNA), based on NEDD4 image intensity (NT = nontargeting) *, P < 0.05 by Welch's two-tailed *T* test. **(C)** TDP-43-Dendra2 half-life in iNeurons with or without NEDD4 depletion. P value determined using sum of squares F test. Error bars in all graphical panels represent standard error of the mean.

MVBs. In both instances, we saw clear colocalization of Rab7 and CD63 (Fig. S4 A), albeit with distinct morphologies. In WT cells with endogenous TDP-43 expression levels, CD63 and Rab7 often appeared partly juxtaposed on small vesicular structures or adjacent foci (Fig. S4 A top row), whereas in TDP-43-GFP–expressing cells, CD63 and Rab7 colocalization was more overlapping, particularly on the vesicular membrane itself (Fig. S4 A second row).

To further examine if the giant Rab7 vesicles were of endolysosomal origin, we examined if CD63, which is also a marker of MVB intraluminal vesicles (Pols and Klumperman, 2009), formed intraluminal "bodies" (puncta and discernible vesicular-

like structures) within Rab7 vesicles in TDP-43-GFP–expressing cells. Besides CD63 signal that colocalized directly with the Rab7 vesicular membrane, we often observed CD63 bodies adjacent to the luminal side of the Rab7 vesicular membrane (Fig. S4 B; top row). Rarely, we observed fully intraluminal CD63 (and Rab7) bodies, very occasionally with a TDP-43 puncta associated (Fig. S4 B, second row). 6% of Rab7 vesicular luminal area was occupied by CD63 bodies on average (Fig. S4, B and C). However, following CHQ treatment, the percentage of Rab7 vesicular luminal area occupied by CD63 bodies increased significantly to 19% (Fig. S4, B and C). This preliminarily suggests that giant Rab7 vesicles are proteolytically active sites where CD63 bodies may

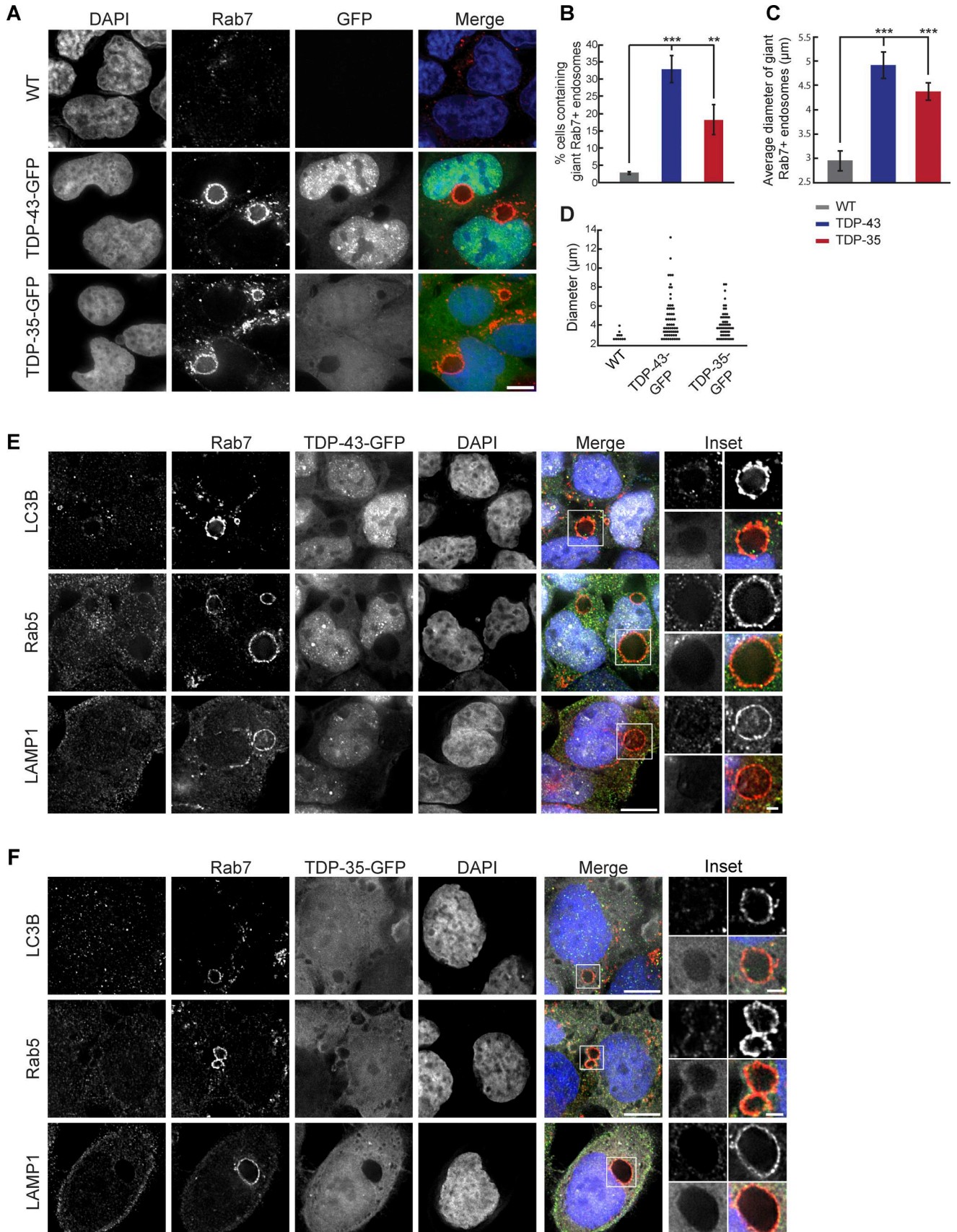

Figure 7. **Mild TDP-43 overexpression causes accumulation of giant Rab7-positive vesicles. (A)** HEK293A WT, HEK-TDP-43-GFP, and HEK-TDP-35-GFP cells were grown in normal conditions before being fixed and immunostained for Rab7. $N$ = 3; scale bar, 10 µm. **(B)** Quantification of the frequency of giant

endosomes in HEK293A WT, HEK-TDP-43-GFP, and HEK-TDP-35-GFP cells. **, P < 0.01; ***, P < 0.001 by one-way analysis of variance with Tukey's post hoc test; N = 3. **(C)** Quantification of the average size of giant endosomes in HEK293A WT, HEK-TDP-43-GFP, and HEK-TDP-35-GFP cells. ***, P < 0.001 by one-way analysis of variance with Tukey's post hoc test; N = 3. **(D)** Distribution of giant Rab7+ve vesicle diameters from WT, HEK-TDP-43-GFP, and HEK-TDP-35-GFP cell lines. N = 3. **(E)** HEK-TDP-43-GFP (pseudocolored white) cells were grown in normal conditions before being fixed and immunostained for Rab5, LC3B, or LAMP1 (pseudocolored green) and then successively immunostained for Rab7 (pseudocolored red). N = 3; scale bar, 10 µm. **(F)** HEK-TDP-35-GFP (pseudocolored white) cells were grown in normal conditions before being fixed and immunostained for Rab5, LC3B, or LAMP1 (pseudocolored green) and then successively immunostained for Rab7 (pseudocolored red). N = 3; scale bar, 10 µm in main panels; 2 µm in insets. Error bars in all graphical panels represent standard error of the mean.

be degraded. CHQ treatment also caused a significant decrease in the size of giant Rab7 vesicles (Fig. S4 D), though their overall number per cell increased dramatically. Why CHQ should impact giant Rab7 vesicle size is unclear but could reflect altered endolysosomal fusion/fission dynamics.

In summary, our giant Rab7-positive vesicular structures appear to resemble MVBs from a compositional perspective, and the presence of intraluminal CD63 bodies, particularly following CHQ treatment. However, the greatly enlarged size of these vesicles and the low levels of intraluminal CD63 signal in untreated cells suggest a significant defect in endolysosomal trafficking, consistent with prior work (Liu et al., 2017; Liu et al., 2020).

### TDP-43 accumulates within giant Rab7-positive vesicles in a NEDD4-dependent manner

Although degradation of overexpressed TDP-43-GFP or TDP-35-GFP cannot rely solely on degradation in giant Rab7 vesicular compartments (only ~20–35% of cells exhibit such organelles; Fig. 7 B), we were curious if TDP-43 may accumulate in such vesicles. While TDP-43-GFP– and TDP-35-GFP–expressing cells did not show striking GFP signal within giant Rab7-positive vesicles, we presumed that giant Rab7 vesicles may be proteolytically active, and therefore, internalized TDP-43/35-GFP may be quickly degraded by resident proteases. Additionally, the lower pH environment in canonical MVBs can impair GFP fluorescence, which is sensitive to acidic pH (Shinoda et al., 2018). To address both these issues, we again used CHQ, which could increase detection of any GFP signal within our Rab7-positive vesicles by increasing GFP fluorescence and impairing resident proteases. Indeed, when we treated TDP-43-GFP and TDP-35-GFP cells with CHQ, the GFP signal within a subset of giant Rab7-positive vesicles became more readily apparent (Fig. 8, A and C; and Fig. S5 A). While some Rab7-positive vesicles exhibited relatively uniform accumulation of TDP-43-GFP (Fig. 8 A, row 2), we observed other cases in which the TDP-43-GFP signal appeared in invagination-like structures around the perimeter of the Rab7-positive membrane (Fig. 8 A, row 3). Additionally, when we transfected TDP-43-GFP and TDP-35-GFP cells with NEDD4-mCherry and then treated the cells with CHQ, we found NEDD4-mCherry colocalized with the TDP-43-GFP and TDP-35-GFP invagination events around the Rab7-positive membrane (Fig. S5 B).

To determine if TDP-43 localized within CD63-labeled ILVs, we also examined TDP-43, CD63, and Rab7 colocalization in TDP-43-GFP–expressing cells ± CHQ. Various TDP-43 cytoplasmic localization phenotypes were observed, including distinct TDP-43 cytoplasmic puncta (Fig. S4 B, row 3 and 6), TDP-43 puncta on

the membrane of Rab7/CD63-positive vesicles (Fig. S4 B, row 5) and TDP-43 puncta that localized with CD63 puncta, or within CD63 vesicles, that together were within giant Rab7 vesicles (Fig. S4 B, row 2 and 4). Notably, CD63-associated/encapsulated TDP-43 puncta, which localized within Rab7 vesicles, increased in the presence of CHQ (Fig. S4 E). Collectively, this suggests that TDP-43 internalization within giant Rab7 vesicles may occur via a mechanism similar to ESCRT-driven invagination events at normal MVBs.

Finally, we examined if NEDD4 knockdown prevented TDP-43 accumulation within giant Rab7-positive vesicles. We found that NEDD4 knockdown significantly decreased the percentage of cells with TDP-43-GFP within Rab7-positive vesicles after CHQ treatment (Fig. 8, B and C). This, combined with our earlier data, suggests that TDP-43 is ubiquitinated by NEDD4, which ultimately targets it for internalization into an MVB-like organelle (canonical or "giant"), presumably aided by the ESCRT complex.

## Discussion

Here, we identified ESCRT and Rsp5/NEDD4 as key facilitators of TDP-43 turnover via our previously identified endolysosomal degradation mechanism. A key question is how TDP-43 interactions with Rsp5/NEDD4 and the ESCRT complex result in such degradation. Based on our data, and prior work discussed below, we hypothesize that TDP-43, possibly aided by E3 ubiquitin ligase adaptor function (e.g., Rim8 in yeast), is ubiquitinated by Rsp5/NEDD4, which facilitates TDP-43 interaction with various ESCRT complex subunits via their ubiquitin binding domains. TDP-43 is then incorporated within MVB intraluminal vesicles via ESCRT-mediated invagination. TDP-43 could be degraded either in MVBs (Krause and Cuervo, 2021) or following trafficking to vacuoles/lysosomes (Fig. 9).

This working model resembles endosomal microautophagy (eMI), which has been described for some cytosolic soluble proteins in human cells, fruit flies, and fission yeast (Schizosaccharomyces pombe), but previously not budding yeast (Saccharomyces cerevisiae) (Liu et al., 2015; Mukherjee et al., 2016; Sahu et al., 2011). eMI usually involves the Hsc70 chaperone binding a KFERQ-like motif within the substrate protein, which then targets it to the MVB via Hsc70 binding of phosphatidylserine residues on the MVB membrane (Morozova et al., 2016). Substrates are then internalized into intraluminal vesicles in a manner reliant on ESCRT-I, II, and III complexes and Vps4 (Mukherjee et al., 2016; Sahu et al., 2011; Tekirdag and Cuervo, 2018). Under starvation conditions, Hsc70 and ESCRT-I become dispensable for eMI (Mejlvang et al., 2018). Consistent with

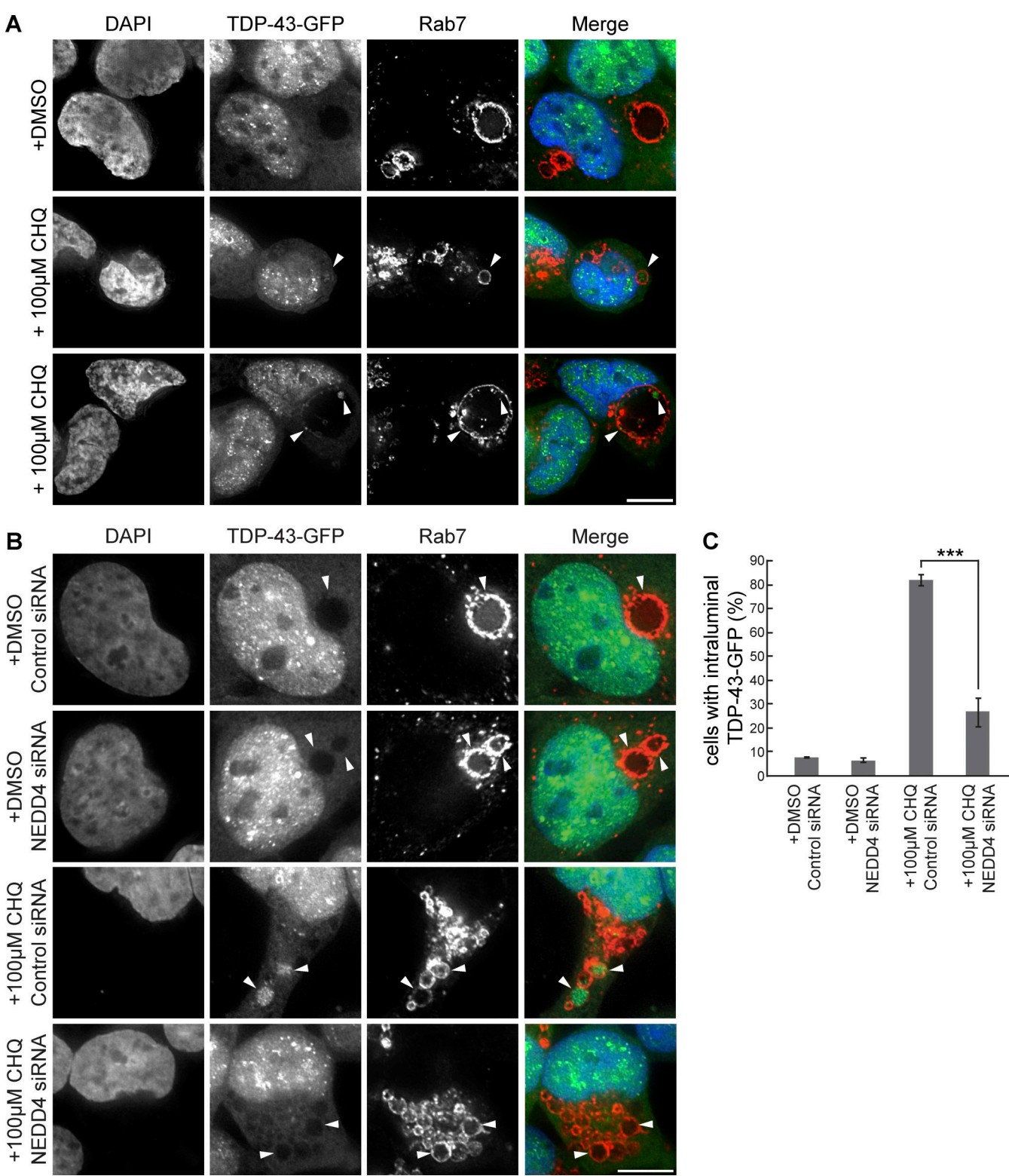

Figure 8. **Giant Rab7-positive vesicles are proteolytically active compartments. (A)** HEK-TDP-43-GFP cells were grown in media containing solvent control (DMSO) or 100 µM CHQ for 18 h before being fixed and immunostained for Rab7. **(B)** N = 3; scale bar, 10 µm (B) HEK-TDP-43-GFP cells were transfected with siRNAs targeting NEDD4 or noncoding control and grown in normal conditions for 48 h. Cells were then treated with solvent control (DMSO) or 100 µM CHQ for 18 h before being fixed and immunostained for Rab7. **(C)** N = 3; scale bar, 10 µm (C) Quantification of data from B showing percentage of cells with enriched TDP-43-GFP signal within Rab7-stained MVBs. ***, P < 0.001 by one-way analysis of variance with Tukey's post hoc test; N = 3. Error bars represent standard error of the mean. Scale bar, 10 µm.

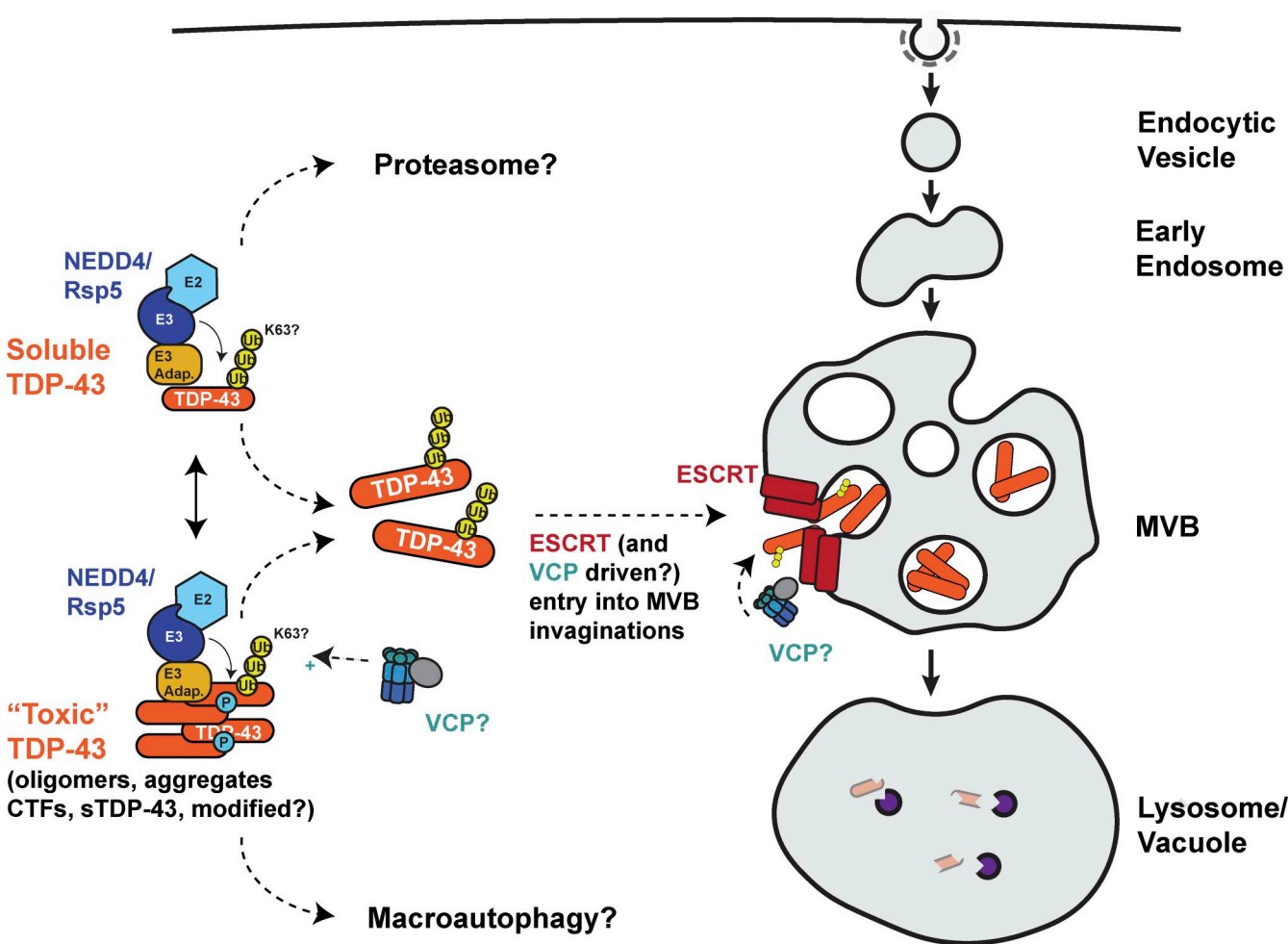

Figure 9. **Potential model of Rsp5/NEDD4 and ESCRT-dependent clearance of cytoplasmic TDP-43.** We speculate that TDP-43 is subject to K63 polyubiquitination by Rsp5/NEDD4, which may promote recognition and remodeling by the "Ub-segregase" chaperone VCP and/or subsequent binding to ESCRT factors via their Ub-binding domains. This may lead to TDP-43 internalization within MVB intraluminal vesicles and subsequent degradation in lysosomes (or MVBs themselves). While our data indicate full-length TDP-43 degrades via our endolysosomal pathway, it remains unclear what, or if, specific TDP-43 physical states (e.g., cleaved/modified forms, aggregates) preferentially degrade via this or other reported TDP-43 clearance mechanisms (e.g., macroautophagy and proteasome).

TDP-43 being a possible eMI target, knockdown of Tsg101 (ESCRT-I) and Vps24 (ESCRT-III) in HeLa cells causes TDP-43 cytoplasmic aggregation, though this was assumed to reflect macroautophagic defects (Filimonenko et al., 2007). TDP-43 also harbors a KFERQ motif and is bound by Hsc70 (Huang et al., 2014). No role for ubiquitination to our knowledge has been described in human eMI. However, in *S. pombe*, transport of cytosolic hydrolases to vacuoles occurs via a macroautophagy-independent, ESCRT, and ubiquitination-dependent mechanism termed the Nbr1-mediated vacuolar targeting pathway (Liu et al., 2015). Specifically, Nbr1, three E3 ubiquitin ligases (two of which are Rsp5 homologs), and a Ub-binding domain within the ESCRT-0 protein Vps27 aid this vacuolar targeting process. Collectively, these studies provide additional support for our model and highlight the importance of assessing the role of Hsc70, TDP-43's KFERQ motif, and ESCRT ubiquitin-binding functions in future work. Additionally, defining the precise nature of the endolysosomal compartments that TDP-43 is targeted to, via further protein/lipid compositional studies and

high-resolution microscopy methods (e.g., electron microscopy), will be key to fully understanding TDP-43 endolysosomal degradation.

Alternative models of TDP-43 endolysosomal clearance merit consideration. KFERQ motifs and Hsc70 binding are also generally required for chaperone-mediated autophagy (CMA), in which unfolded proteins are directly translocated across the lysosomal membrane through a LAMP2A translocation pore. While clearance of CTFs of TDP-43 by CMA has been reported (Ormeño et al., 2020), this pathway is absent in yeast and makes no use of ESCRT components in human cells; thus, it is unlikely to be the turnover mechanism described in our studies (Liu et al., 2017; Liu et al., 2020). In contrast, the ESCRT complex, aided by Rsp5 and the Ssh4 adaptor, can also drive vacuole membrane invagination and degradation of vacuolar membrane proteins in yeast (Zhu et al., 2017). However, *ssh4Δ* yeast cells do not show altered TDP-43 levels (Fig. S1), despite colocalizing in TDP-43 aggregates (Fig. 3 D). Also, TDP-43 accumulation within Rab7-positive, LAMP1-negative organelles (Fig. 7 and Fig. S4) is more

consistent with MVB than lysosomal (vacuolar equivalent) targeting.

Endolysosomal dysfunction may play a key role in the ALS disease mechanism and possibly contributes to cytoplasmic TDP-43 pathology. Several familial ALS genes (*ALS2* [Hadano et al., 2001; Kunita et al., 2004], *FIG4* [Chow et al., 2009], and *CHMP2B* [Skibinski et al., 2005; Parkinson et al., 2006]) have clear endolysosomal functions, with *ALS2* being a Rab5 guanine exchange factor (GEF), *FIG4* being involved in PI(3)P synthesis (critical for early endosome and autophagosome biogenesis), and *CHMP2B* being an ESCRT-III complex factor. Another familial ALS gene and kinase, *TBK1* (Cirulli et al., 2015; Freischmidt et al., 2015) phosphorylates endosomal proteins (including Rab7), may modulate Rab7 GTPase activity (Heo et al., 2018), colocalizes with early endosome markers (Chau et al., 2015), and leads to severe defects in endolysosomal trafficking and lysosomal acidification when absent in motor neurons (Hao et al., 2021, *Preprint*). Also, the most common familial ALS mutant gene, *C9ORF72*, harbors a DENN domain, which is associated with GEFs for Rab GTPases (Farg et al., 2014; Levine et al., 2013). C9orf72 also colocalizes with early endosome and MVB markers, including in ALS patient motor neurons, and its depletion impairs endolysosomal flux (Farg et al., 2014). Finally, partial impairment of *TBK1* activity via an ALS-associated R228H point mutation, combined with $(GA)_{100}$ peptide expression (*C9ORF72* dipeptide mimic), showed increased TDP-43 pathology and defects in endolysosomal flux (Shao et al., 2022). Notably, *TBK1* and *C9ORF72* have also been linked to macroautophagic activities (Heo et al., 2018; Nassif et al., 2017; Pilli et al., 2012; Webster et al., 2016). Regardless, these studies indicate that endolysosomal dysfunction may lead to cytoplasmic TDP-43 pathology, likely due to impaired TDP-43 clearance.

Additionally, exceeding a certain cytoplasmic threshold of TDP-43 pathology may inhibit endolysosomal flux. Overexpression of TDP-43 alleles increasingly prone to cytoplasmic mislocalization and aggregation correlates closely with endocytosis rate defects (Liu et al., 2017; Liu et al., 2020). Cytoplasmic TDP-43 aggregation causes dysregulation and lowered abundance of Hsc70-4 at neuromuscular junctions, which drives synaptic vesicle recycling, and endocytosis-dependent processes (Coyne et al., 2017). This could underlie ALS motor neuron defects, such as glutamate excitotoxicity, where elevated concentrations of glutamate receptors at synapses are observed (Shi et al., 2018). Additionally, proximity labeling data specific to aggregation-prone TDP-43 CTFs exhibited strong TDP-43 interactions with vesicular transport proteins, including endolysosomal and Golgi–ER-related pathways (Chou et al., 2018). Here, and in our prior work, microscopy confirms that numerous endolysosomal trafficking proteins are sequestered in yeast and human TDP-43 cytoplasmic aggregates (Liu et al., 2017). Thus, while endolysosomal clearance of TDP-43 may be a common response to keep cytoplasmic TDP-43 pathology in check, this pathway might be overwhelmed by accumulation of TDP-43 above a certain cytoplasmic threshold, leading to sequestration of key endolysosomal factors in TDP-43 aggregates. This could cause a positive feedback loop of further impairment of TDP-43 clearance and greater sequestration of endolysosomal factors,

possibly resulting in compensatory clearance mechanisms being induced (e.g., macroautophagy) as suggested previously (Hao et al., 2021, *Preprint*).

Giant Rab7-positive MVB-like structures in our HEK293A TDP-43-GFP and TDP-35-GFP models further supports a defective endolysosomal trafficking process related to TDP-43 pathology. Other giant vesicular structures of presumed endolysosomal origin have been observed in ALS-relevant contexts. These include *FIG4* mutant mouse fibroblasts, where giant LAMP-2 positive organelles were seen in ~40% of cells (versus 5% in WT) (Chow et al., 2007). In "wobbler" mutant mice, which develop a progressive ALS-like motor neuron degeneration phenotype due to homozygous Vps54 Q967L mutations (impairs endosome-Golgi trafficking), giant endosomal vesicles were observed in motor neurons, which were Rab7-positive and mostly LC3-negative, mimicking our observations. Amyloid-precursor protein (APP; implicated in Alzheimer's) also colocalized in these structures. These giant endosomal vesicles were also observed in almost half of spinal motor cord neurons from sporadic ALS patients (Palmisano et al., 2011). Finally, in the cortex cells of mice overexpressing a $(GA)_{100}$ peptide, which also exhibited TDP-43 pathology, both EEA1 (early endosome marker) and Rab7 (MVB marker) positive giant endosomes were observed (Shao et al., 2022). The frequency of these vesicles increased further in a *TBK1* hypomorphic background, consistent with the previously discussed endolysosomal trafficking defects of *TBK1* mutants. While we have not examined LAMP2, EEA1, or APP here, these studies all suggest an endolysosomal trafficking defect in various ALS models and ALS patient samples. Consistent with the notion that TDP-43 cytoplasmic accumulation may be involved, giant Rab7-positive vesicles are observed in >30% of HEK293A cells where endogenous TDP-43 and TDP-43-GFP are co-expressed (twofold WT levels), whereas WT HEK293A cells display this in <5% of cells (Fig. 7 B). Determining whether simple sequestration of endolysosomal factors within TDP-43 cytoplasmic aggregates or other TDP-43–mediated inhibitory mechanisms impact endolysosomal trafficking, particularly giant Rab7-positive vesicle formation, are areas of future interest.

Approximately 600–700 E3 ubiquitin ligases exist in human cells, several of which (Parkin, Cul2, RNF4, Znf179, and SCF) have been previously linked via candidate-based studies to TDP-43 ubiquitination. Overexpression of both TDP-43 and Parkin in rat brains led to colocalization of both proteins in cortical neurons, though Parkin expression actually increased cytoplasmic TDP-43 mislocalization. Additionally, while Parkin ubiquitinated TDP-43 *in vitro* with both K48- and K63-linked Ub chains, *in vivo* data did not reveal any impact on TDP-43 ubiquitination nor TDP-43 protein levels (Hebron et al., 2013). The VHL/Cul2 E3 ubiquitin ligase complex ubiquitinates the 35-kDa CTF TDP-43 *in vitro*, colocalizes with TDP-43 inclusions in HEK293A and in ALS patient oligodendrocytes, and facilitates proteasomal-dependent turnover of CTFs *in vivo* based on genetic studies (Uchida et al., 2016). No impact of VHL/Cul2 on full-length TDP-43 levels was observed. RNF4 binds TDP-43 when both proteins are overexpressed. Under heat stress, endogenous TDP-43 exhibits *in vivo* ubiquitination that decreases following RNF4

knockdown, though again no impact on TDP-43 levels was observed (Keiten-Schmitz et al., 2020). In mice and N2a cells, Znf179 binds and ubiquitinates TDP-43 in vivo and in vitro, and Znf1790 overexpression aids turnover of endogenous or overexpressed TDP-43. Cytoplasmic TDP-43 aggregates also increase in Znf179 KO mice brain tissue (Lee et al., 2018). Finally, in HEK293 cells, the SCF E3 ubiquitin ligase complex binds TDP-43 via the cyclin F subunit and adds K48 ubiquitin to TDP-43 in vitro (Rayner et al., 2022). Furthermore, cyclin F overexpression drives TDP-43 ubiquitination and reduces TDP-43 levels in HEK293 cells but also increases cell death (Rayner et al., 2024).

To our knowledge, Rsp5/NEDD4 is the first E3 ubiquitin ligase identified via unbiased methods to both promote TDP-43 turnover (in yeast, HEK293A, and iNeurons) and reduce TDP-43 toxicity (yeast and HEK293A). TDP-43 was also identified as a putative NEDD4 target in a global proteomics study in neural crest cells, with a ubiquitination at K145 within the RRM1 domain (Lohraseb et al., 2022). K63 ubiquitination of TDP-43 under arsenite stress also decreases following chemical inhibition of HECT E3 ubiquitin ligases (Pan et al., 2025). Besides demonstrating that Rsp5/NEDD4 regulates TDP-43 levels, stability, aggregation, toxicity, and ubiquitination status, NEDD4 is also the first E3 ubiquitin ligase shown to impact TDP-43 targeting to a specific organelle (giant Rab7-positive vesicles); we suspect NEDD4 likely targets TDP-43 to canonical MVBs also, but our microscopy data lacked sufficient resolution to resolve this. Regardless, NEDD4 is a consequential regulator of TDP-43 proteostasis by endolysosomal degradation. However, we cannot fully rule out that NEDD4 may promote TDP-43 clearance by other means, nor that TDP-43 clearance may be regulated by multiple E3 ubiquitin ligases and adaptors.

Like TDP-43, α-synuclein, a Lewy body component present in Parkinson's disease, also degrades in yeast and human cell culture models via an autophagy- and proteasomal-independent mechanism. This too relies on ESCRT and Rsp5/NEDD4 function, with NEDD4-mediated K63 ubiquitination of α-synuclein having been demonstrated (Tardiff et al., 2013; Tofaris et al., 2011). Fly and rat models of Parkinson's, featuring ectopic α-synuclein expression, show suppression and enhancement of α-synuclein pathological effects with NEDD4 overexpression or knockdown, respectively (Davies et al., 2014). Thus, many proteins besides TDP-43 and α-synuclein are likely cleared via an Rsp5/NEDD4- and ESCRT-regulated endolysosomal pathway.

Distinct TDP-43 protein isoforms arising from ALS-associated mutations, alternative splicing, or posttranslational cleavage exhibit divergent protein half-lives from WT TDP-43 (Berning and Walker, 2019; Brower et al., 2013; Flores et al., 2019; Pesiridis et al., 2011; Scotter et al., 2014; Watanabe et al., 2013), implying distinct clearance pathways are likely used. Thus, TDP-43 isoform, misfolding, aggregation, and modification status may be determining factors for endolysosomal clearance pathway use. This could be regulated by VCP/p97 (Fig. 9), a Ub-segregase and AAA-ATPase chaperone, which also facilitates TDP-43 clearance (Liu et al., 2020) and which can facilitate entry of ubiquitinated substrates into MVBs (Burana et al., 2016; Jang et al., 2019; Kirchner et al., 2013; Ritz et al., 2011). Addressing how distinct isoforms of TDP-43 are degraded is an important area of future research, as it may allow selective turnover of critical disease-associated isoforms while preserving functional TDP-43 species.

Finally, it is important to determine what effect (if any) NEDD4 and the endolysosomal TDP-43 degradation mechanism has on TDP-43 secretion. In various contexts, MVBs can fuse with the plasma membrane and release their ILVs as exosomes (Baixauli et al., 2014; Ponpuak et al., 2015). Indeed, neurodegenerative disease-associated proteins, including Tau and α-synuclein, are thought to be secreted following sequestration in endosomal/MVB compartments (Caballero et al., 2021; Wu et al., 2023), in a manner influenced by protein conformation and cellular aging. Acetylated and oligomeric forms of Tau are preferentially targeted to MVBs via eMI, a portion of which is secreted (Caballero et al., 2021). Furthermore, eMI-targeted proteins targeted show increased secretion at the expense of degradation in aged mouse fibroblast cells (Krause et al., 2022). This switch in protein fate may depend on aged-dependent increases in Hsc70 glycation, which increase recruitment of the exocyst complex and RalA (exocyst regulatory GTPase) to the MVB membrane. Notably, the exocyst–RalA complex inhibits eMI degradation, perhaps reflecting a role in promoting MVB plasma membrane fusion (Krause et al., 2022). Tau, α-synuclein, and TDP-43 secretion can also occur via an unconventional secretion pathway, independent of ESCRT and MVBs, termed the misfolding-associated protein secretion (MAPS) pathway (Fontaine et al., 2016; Lee et al., 2016; Lee et al., 2022; Xu et al., 2018). MAPS involves translocation of substrates into the lumen of a perinuclear secretory compartment, key positive effectors of which include the deubiquitinase/chaperone USP19, HSC70, and an associated HSP40 cofactor, DNAJC5/CSPα. Interestingly, DNAJC5 overexpression promotes targeting of misfolded proteins by both MAPS and eMI, with differing domains and interacting partners of DNAJC5 required for these roles (Lee et al., 2022; Lee et al., 2023), suggesting DNAJC5 may be an important regulator of protein fate. Collectively, these observations indicate the importance of determining how TDP-43 folding and modification status, aging, eMI/MAPS regulators, and regulators of other proteostatic mechanisms (Keating et al., 2022) ultimately influence TDP-43 degradation and secretion, which likely have important implications for ALS disease progression.

## Materials and methods

### Yeast strains and growth conditions

Untransformed yeast strains were cultured at 30°C with yeast extract-peptone-dextrose medium. Strains transformed with plasmids were grown in synthetic defined (SD) medium with appropriate nutrients for plasmid selection. In liquid culture, strains expressing GAL1-driven TDP-43 from a plasmid (or vector control) were initially cultured overnight in 2% sucrose and then back diluted to an optical density at 600 nm ($OD_{600}$) of 0.1–0.15 in 0.25% galactose and 1.75% sucrose medium, followed by growth at 30°C to mid-logarithmic growth phase ($OD_{600}$, 0.4–0.7). Transformations were performed by a standard lithium acetate method. All yeast strains, plasmids, and antibodies used in the study are listed in Table S3.

## Yeast serial dilution growth assays

Yeast strains transformed with plasmids expressing TDP-43 or an empty vector were grown overnight as described above, back diluted to an $OD_{600}$ of 0.2, serially diluted (1:5 dilutions), and spotted onto identical agar media. All images are representative of those from a minimum of three biological replicates imaged on the same day.

## Dot-blot genetic screen

To identify genes that affect TDP-43 protein levels, we developed a yeast genetic screening method utilizing high throughput yeast colony dot-blots. The yeast gene deletion library (Winzeler et al., 1999) was transformed via a high-throughput transformation approach (Gietz and Schiestl, 2007) with a single-copy plasmid (pRB109) encoding TDP-43-YFP under the control of a galactose-regulatable promoter. In total, we successfully transformed and screened 3,663 strains in >3 biological replicate dot-blot screens, representing 74% of the nonessential gene deletion collection (4,944 strains total). An outgrowth of transformants was then performed in 96-well suspension culture plates for 4 days following transformation in SD media lacking uracil and supplemented with 2% glucose. These yeast strains were then pinned in 384-well format onto agar plates containing the appropriate selection with 2% sucrose as the carbon source using a Singer ROTOR HDA. After growing for 48 h at 30°C, the yeast colonies were then re-pinned in 384-well format onto agar plates that had been overlaid with nitrocellulose. These nitrocellulose-coated agar plates contained appropriate plasmid selection and 1.75% sucrose and 0.25% galactose to induce the expression of TDP-43-YFP. After growth for 8 h at 30°C, the nitrocellulose was removed from the agar plate and placed onto filter paper that had been soaked in lysis buffer (350 mM NaOH, 0.1% SDS, and 1% 2-mercaptoethanol) for 15 min to lyse the cells *in situ*. Blots were then washed three times with 1× TBS + 0.1% Tween-20 to remove cell debris. Blots were then blocked using 5% nonfat dry milk for 45 min at room temperature and incubated with primary antibodies: anti-GFP (ab290; Abcam, dilution 1:5,000) and anti-GAPDH (MA5-15738; Invitrogen, dilution 1:5,000) for 16 h at 4°C. Blots were then washed three times with 1× TBS + 0.1% Tween-20 for 5 min each. Blots were incubated with secondary antibodies: IRDye 800CW goat anti-rabbit (925-32211; LI-COR, dilution 1:20,000) and IRDye 680RD donkey anti-mouse (926-68072; LI-COR, dilution 1:20,000) at room temperature for 1 h. Blots were then washed four times with 1× TBS + 0.1% Tween-20 for 5 min each. Blots were then kept in 1× PBS until being imaged using an LI-COR Odyssey infrared imaging system.

## Western blotting

Western blotting was conducted using standard yeast alkaline lysis and gel electrophoresis protocols. Primary antibodies were as follows: anti-GFP (QAB10298; enQuireBio, dilution 1:5,000), anti-GAPDH (MA5-15738; Invitrogen, dilution 1:5,000), anti-αTubulin (66031; Proteintech, dilution 1:5,000), anti-TDP-43 (10782-2-AP; Proteintech, dilution 1:5,000), anti-NEDD4 (21698-1-AP; Proteintech, dilution 1:1,000), anti-pan Ubiquitin (sc-8017; Santa Cruz, dilution 1:1,000), K63 Ubiquitin (ab179434; Abcam, dilution 1:500), anti-K48 Ubiquitin (D905; Cell Signaling Technology, dilution 1:500), and anti-mCherry (ab167453; Abcam, dilution 1:1,000). The results of the quantification of key western blotting datasets from throughout the paper are listed in Table S2.

## Cycloheximide chase assays

For TDP-43 protein stability assays in yeast, TDP-43 expression was halted by the addition of cycloheximide at 0.2 mg/ml to the growth media. Protein lysates were obtained via a standard NaOH method and examined via western blot. For TDP-43-GFP protein stability assays in HEK293A cells, TDP-43-GFP expression was halted by the addition of cycloheximide at 0.3 mg/ml to the growth media. Protein lysates were generated by resuspending cell pellets in a standard SDS loading buffer followed by incubation at 95°C for 10 min and examined via western blot.

## Yeast fluorescence microscopy

Mid-log live yeast cells ($OD_{600}$ 0.4–0.7) were rapidly examined at room temperature on glass slides (cat: 1301; Globe Scientific) with 1.5 coverslips (cat: 48366-227; VWR) using a DeltaVision Elite microscope with a ×100 oil-immersion (cat: 16245; Cargille, 1.515) objective (NA 1.40) and 15-bit PCO Edge sCMOS camera. Z-stack data (10 slices, 0.4 µm each) were collected. Images were subject to deconvolution using standard parameters (enhanced ratio, 10 cycles, and medium filtering) using SoftWoRx 7.0.0 (DeltaVision software), followed by maximum intensity Z-stack projection. Insight SSI LED light source (475/28 nm for GFP and 575/25 nm for mCherry) was used for excitation, with a Quad-mCh dichroic filter, and filters (Chroma) of 523/36 nm (GFP) and 623/60 nm (mCherry) were used for emission.

All image analysis was done using Fiji software (Schindelin et al., 2019). TDP-43 foci colocalization with different proteins and TDP-43 foci per cell was scored manually, in a blind manner, in a minimum of 50 cells (in which each cell may contain multiple foci). Cell and foci counts for all microscopy data (yeast and human) are listed in Table S2. All shown images are representative of >3 biological replicates, in which each biological replicate involves analysis of a minimum of 50 cells.

## Immunoprecipitation interaction analyses

For immunoprecipitation analyses in yeast cells, 10 ODs of mid-logarithmic growth phase yeast were pelleted by centrifugation at 13,000 × *g* at 4°C and resuspended in 1 ml of radioimmuno-precipitation (RIPA) assay cell lysis buffer. 50% volume of disruption beads was added, and the cells were vortexed for 2 min before being incubated on ice for 3 min. This process was repeated twice. The cells were centrifuged at 17,000 × *g* for 10 min at 4°C to separate unlysed cells and insoluble cell debris. Supernatants were applied to GFP-Trap magnetic agarose beads (gtma, ChromoTek) and nutated for 2 h at 4°C. Beads were then washed four times with wash buffer before being resuspended in standard SDS loading buffer, boiled at 95°C for 10 min, and examined via western blot.

For NEDD4 immunoprecipitation analysis in HEK293A cells, 10-cm dishes of cells stably expressing TDP-43-GFP were transfected with NEDD4-mCherry (pRB334) or mock control and grown to 80–90% confluence over 2 days. Cells were collected via

centrifugation and washed in cold 1× DPBS twice before being resuspended in RIPA assay cell lysis buffer. The cells were centrifuged at 17,000 × $g$ for 10 min at 4°C to separate insoluble cell debris. Supernatants were applied to RFP-Trap magnetic agarose beads (rtma, ChromoTek) and nutated for 2 h at 4°C. Beads were then washed four times with wash buffer before being resuspended in standard SDS loading buffer, boiled at 95°C for 10 min, and examined via western blot. For TDP-43 immunoprecipitation analysis in WT HEK293A cells, the same process was followed as above except that protein-A Dynabeads (10001D; Invitrogen) were conjugated to αTDP-43 (671834; R&D), and Pierce lysis buffer (50 mM tris, 2 mM EDTA, 300 mM NaCl, 10% glycerol, and protease inhibitor cocktail [Roche Diagnostics]) was used.

## Culture of HEK293A cells and HEK293A cells expressing TDP-43-GFP or TDP-35-GFP

TDP-43-GFP and a 35-kDa C-terminal TDP-43 fragment (TDP-35-GFP; amino acids 90–414), a cytoplasmic aggregation-prone allele mimicking TDP-43 cleavage by caspases (Zhang et al., 2009), were stably integrated by retroviral means (pQCXIH vector, hygromycin selection) into HEK293A cells by retroviral means (Liu et al., 2017). Resulting expression of TDP-43-GFP and TDP-35-GFP are comparable with the level of endogenous TDP-43, meaning both lines have about double the level of TDP-43 species (endogenous and GFP-tagged versions) when compared with HEK293A WT cells. HEK293A WT, HEK-TDP-43-GFP, and HEK-TDP-35-GFP cells were cultured at 37°C in Dulbecco modified Eagle medium (10-013-CV; Corning) with 10% fetal bovine serum (26140079; Gibco) and 50 U/ml penicillin and 50 μg/ml streptomycin (15140-122; Gibco). Plasmid transfections were performed using Lipofectamine 3000 according to the manufacturer's protocol (L3000008; Thermo Fisher Scientific). siRNA transfections were performed using Lipofectamine RNAiMAX according to the manufacturer's protocol (13778100; Thermo Fisher Scientific). Lines were verified to be mycoplasma-free on a monthly basis.

## Cell line proliferation assessment

HEK293A, HEK-TDP-43-GFP, and HEK-TDP-35-GFP cells were seeded in 6-well plates at $1 \times 10^5$ cells per well. Cells were then transfected with 2.5 μg NEDD4-mCherry or VPS4A$^{E228Q}$ plasmid, or 25 pmol NEDD4 siRNA, or appropriate controls and incubated for 48 h. Cell numbers were manually counted using a hemocytometer and normalized to the WT control wells.

## Ubiquitination assay via immunoprecipitation

To assess TDP-43-GFP ubiquitination in HEK-TDP-43-GFP cells, 10-cm dishes of cells were transfected with a NEDD4-mCherry expression plasmid (pRB334), siRNAs targeting NEDD4, or mock control and grown to 80–90% confluence over 3 days. Cells were collected via centrifugation and washed in cold 1× DPBS twice before being resuspended in urea lysis buffer (8M urea, 50 mM Tris-HCl, pH 8.0, 75 mM NaCl, 100 mM N-ethylmaleimide, 1 mM β-glycerophosphate, 1 mM NaF, 1 mM NaV, and EDTA-free protease inhibitor cocktail [Roche Diagnostics]). Lysates were applied to GFP-Trap magnetic agarose beads and nutated for 2 h at 4°C. Beads were then washed four times with urea lysis buffer before being resuspended in standard SDS loading buffer, boiled at 95°C for 10 min, and examined via western blot.

To assess endogenous TDP-43 ubiquitination in WT HEK293A cells, 15-cm dishes were transfected with an HA-Ub plasmid (pRB506) and NEDD4-mCherry (pRB334) and collected around 80% confluency over 2 days. Cells were collected and washed as above and resuspended in lysis buffer (10 mM Tris-HCl, pH 7.5, 150 mM NaCl, 0.5 mM EDTA, 0.1% NP-40, 10 uM MG132, 50 mM N-ethylmaleimide, and EDTA-free protease inhibitor cocktail [Roche Diagnostics]). Lysates were applied to Protein A Dynabeads (10001D; Invitrogen) conjugated to TDP-43 antibody (671834; R&D) and nutated overnight at 4°C. Beads were washed three times in lysis buffer, and samples prepared for western blot analysis as above.

## Cell line immunofluorescence

Cells were cultured for 24 h in 8-well chamber slides (80821; ibidi), fixed with 4% paraformaldehyde for 15 min, and permeabilized with 0.1% Triton X-100 for 15 min. Cells were incubated with indicated primary antibodies overnight at 4°C after blocking with 5% bovine serum albumin. Cells were then incubated with Alexa Fluor–conjugated secondary antibody (Thermo Fisher Scientific, 1:1,000 dilution) for 1 h at room temperature. After three washes with PBS, cells were stained with 0.5 μg/ml DAPI (P36931; Thermo Fisher Scientific) and imaged using a DeltaVision Elite microscope as described for yeast, except 20–25 0.3-μm slices were collected per Z-stack. Data in Fig. S4, A and B were imaged on an Olympus IX83 wide-field deconvolution microscope using a 60× oil immersion (F30CC; Olympus) objective (NA 1.42) and 16-bit ORCA-FLASH4.0 LT plus SCMOS camera (Hamamatsu). Z-stack data (30–35 slices, 0.3 μm each) were collected. Images were subject to deconvolution using five iterations of the Advanced Maximum Likelihood deconvolution algorithm using cellSens Dimension 4.2.1 software (Olympus). Primary antibodies used for the immunofluorescence were Rab7 (ab137029; Abcam, dilution: 1:200), CD63 (H5C6; Developmental Studies Hybridoma Bank, dilution 1:100), Rab5 (ab109534; Abcam, dilution: 1:200), LC3B (3868S; Cell Signaling, dilution 1:500), LAMP1 (21997-1-AP; Proteintech, dilution 1:200), and TDP-43 (10782-2-AP; Proteintech; dilution 1:200). All shown images are representative of 3 biological replicates, except for Fig. 8 B, where images shown are representative of 2 biological replicates. All HEK293 cell data were subject to quantification using Fiji (Schindelin et al., 2019). Colocalization analyses were performed by thresholding single z-plane images where objects of interest in 2 distinct channels were in focus, followed by thresholding and masking to generate channel-specific regions of interest (ROIs). Overlap of ROIs between channels was normalized to the total number of ROIs in the channel of primary interest (e.g., fraction of Rab7 ROIs that overlap with CD63 ROIs, divided by total Rab7 ROIs). Rab7 vesicle diameter measurements were conducted manually in >100 WT, TDP-43-GFP, and TDP-35-GFP–expressing cells using the line tool. Percentage of cells with giant Rab7 vesicles (± intraluminal TDP-43) were scored manually in a blinded manner. For Rab7 vesicular volume and CD63 intraluminal occupancy, 3 consecutive Z slices (0.3 μm

thickness), focused on the center of most Rab7 vesicles in the field of view, were max-Z-projected. Rab7 vesicle outlines were then manually drawn blinded to other channel phenotypes. The resulting ROIs were then transferred to the CD63 channel image, and total CD63 area within the ROI was determined based on a thresholding above background signal levels determined from non-CD63 staining controls. CD63 area measurements were then divided by Rab7 ROI area measurements.

## Generation and maintenance of iPSCs

Fibroblasts isolated from control patients via skin punch followed a standard approach as outlined by Tank et al. (2018). Fibroblasts were reprogrammed into iPSCs via transfection with episomal vectors encoding seven reprogramming factors (Oct4, Sox2, Nanog, Lin28, L-Myc, Klf4, and SV40LT [Yu et al., 2011]). Manufacturer protocols provided with the Episomal iPSC reprogramming Vector kit (Invitrogen/Thermo Fisher Scientific) were followed with minor deviations as outlined elsewhere (Tank et al., 2018). iPSC lines were cultured in Essential 8 (E8) media (A1517001; Gibco) on vitronectin (A14700; Gibco)-coated plates. Cells were passaged every 5–6 days using 0.5 mM EDTA (E7889; Sigma-Aldrich) dissolved in PBS followed by gentle trituration in E8 media. Lines were verified to be mycoplasma-free on a monthly basis.

## iNeuron generation and differentiation

iPSCs were integrated with a Ngn1 and Ngn2 cassette (pUCM-CLYBL-NGN1-2-RFP; gift from Dr. Michael Ward, NIH, Bethesda, MD, USA) at the CLYBL safe harbor locus using TALENS as outlined in detail elsewhere (Weskamp et al., 2020; Chua et al., 2022). Briefly, iPSCs were seeded on vitronectin-coated 6-well plates in E8 media (A1517001; Gibco) with ROCK inhibitor (BDB562822; Thermo Fisher Scientific) and changed into fresh E8 media the following morning. 30 min before transfection, when confluency was 50–70%, cells were changed into mTESR-1 media (85850; Cell Technologies) and then transfected with 2.5 µg of donor DNA and 1.25 µg of each targeting construct using Lipofectamine Stem (STEM00003; Invitrogen), following the manufacturer's instructions. Cells underwent daily E8 media changes and were screened for red fluorescence. When partially positive colonies reached 100–500 cells, they were carefully scraped/aspirated using a P200 pipet tip and transferred to a new vitronectin-coated dish. This enrichment process was repeated until a 100% fluorescent colony was identified. This was then relocated to a new dish and expanded for future use.

Differentiation proceeded as follows. Day 0: iPSCs were washed in PBS and incubated in prewarmed accutase (A6964; Sigma-Aldrich) at 37°C for 8 min. Four volumes of E8 media were added to the plate; cells were collected at 200 × g for 5 min. Media was aspirated, and the pellet was resuspended in 1 ml E8 media. Cells were plated at 20,000 cells/ml in E8 media with ROCK inhibitor and incubated at 37°C overnight. Day 1: Media was changed to N2 media (1× N2 Supplement [17502-048; Gibco], 1× NEAA Supplement [11140-050; Gibco], 10 ng/ml BDNF [450-02; PeproTech], 10 ng/ml NT3 [450-03; PeproTech], 0.2 µg/ml laminin [L2020; Sigma-Aldrich], and 2 mg/ml doxycycline [D3447; Sigma-Aldrich] in E8 media). Day 2: Media was changed

to transition media (1× N2 Supplement, 1× NEAA Supplement, 10 ng/ml BDNF, 10 ng/ml NT3, 0.2 µg/ml laminin, and 2 mg/ml doxycycline in half E8 media and half DMEM F12 [11320-033, Gibco]). Day 3: Media was changed to B27 media (1× B27 Supplement [17504-044; Gibco], 1× GlutaMAX Supplement [35050-061; Gibco], 10 ng/ml BDNF, 10 ng/ml NT3, 0.2 µg/ml laminin, 2 mg/ml doxycycline, and 1× Culture One [A33202-01; Gibco] in Neurobasal-A [12349-015; Gibco]). Day 6: An equal volume of B27 media without Culture One was added to each well. Day 9–21. All cultures underwent a half-media change every 3 days in fresh B27 media.

## iNeuron optical pulse labeling microscopy

Neurons were imaged as (Barmada et al., 2015; Malik et al., 2018; Weskamp et al., 2020) using a Nikon Eclipse Ti inverted microscope with PerfectFocus3a 20× objective lens and an Andor iXon3 897 EMCCD camera. A Lambda 4-2-1 multi-LED combiner with 405, 488, 560, and 647 LEDs (Sutter) was used to illuminate samples. For photoconversion, iNeurons were cultured in BrainPhys Imaging media (05796; Stem Cell Technologies) for 24 h prior to illuminating 405 nm light (100 W) for 20 s, timed to occur immediately after the first imaging session. iNeurons were then imaged by automated microscopy at 4–12-h intervals in the GFP and RFP channels. TDP-43-Dendra2 half-life was calculated as described (Flores et al., 2019). Briefly, single-cell RFP intensities were normalized to the values measured immediately after photoconversion. Only cells that lived the entire measurement period (>48 h) were included in the analysis, and cells displaying negative half-lives were removed. Single-cell half-life measurements were averaged for each time point before determining first-order exponential decay kinetics through Prism. Statistical differences in measured half-life values were determined using the sum of squares F test. Custom ImageJ/Fiji macros and Python scripts were used to identify neurons and draw cellular ROIs based upon size, morphology, and fluorescence intensity. Custom Python scripts were used to track ROIs over time to measure Dendra2 green and red fluorescence intensity over time.

## NEDD4 shRNA lentiviral generation

Glycerol stock from the pGIPZ shRNA library targeting NEDD4 (V2LHS_254872) or nontargeting control was purchased from Horizon Discovery. Plasmid DNA was amplified using a Midi kit (QIAGEN) and used to create lentiviral particles through the University of Michigan Vector Core facility. On DIV3, iNeurons were inoculated with 10 µl of lentiviral supernatant/well of a 96-w plate (200 µl total volume) and cultured for 5 days prior to evaluation of TDP-43-Dendra2 half-life by optical pulse labeling.

## Statistical procedures

Graphs and statistical analyses were determined using Graph-Pad Prism Versions 9–10 or Microsoft Excel by Welch's t test, Mann–Whitney U test, ordinary one-way ANOVA with Tukey's post hoc test, ordinary two-way ANOVA with Tukey's post hoc test, or sum of squares F test. * = P < 0.05; ** = P < 0.01; *** = P < 0.001. We used the Shapiro–Wilks test to determine normality in datasets. If data were determined to be normally distributed, we

used parametric analyses (ANOVA, Welch's t test). If data were determined to be not normally distributed, we used nonparametric tests (Mann–Whitney U Test). Error bars represent the standard error of the mean.

## Online supplemental material

Fig S1 shows the impact of Rsp5 adaptor proteins on steady-state TDP-43-YFP levels in yeast. Fig S2 provides evidence that endogenous TDP-43 in HEK293A cells is subject to increased levels of ubiquitination, including K63-linked ubiquitination, following NEDD4 overexpression. Fig S3 reveals endogenous protein interactions between TDP-43, Vps4A, and NEDD4 and the impact of CHQ or NEDD4 depletion on said interactions. Fig S4 examines the composition (Rab7, CD63, and TDP-43) and morphology of the giant Rab7-positive vesicles observed in TDP-43-GFP–expressing HEK293A cells. Fig S5 examines targeting of TDP-43-GFP and TDP-35-GFP to/within giant Rab7-positive vesicles following CHQ treatment or NEDD4 overexpression. Table S1 provides raw dot-blot screen data. Table S2 summarizes all quantitative data and statistical tests within the study. Table S3 summarizes yeast strains, human cell lines, plasmids, antibodies, and siRNAs used in the study.

## Data availability

The data are available from the corresponding author upon reasonable request.

## Acknowledgments

We gratefully acknowledge support from Matt Kaplan and the Functional Genomics Core at the University of Arizona for their help with our genetic screen and the University of Michigan Biomedical Research Vector Core facility. We thank Buchan lab members for their comments on the manuscript. Sincere thanks to Dr. Ralf Braun (Danube Private University, Krems an der Donau, Austria) for sharing of the NEDD4-mCh plasmid and general advice and Nikita Fernandes for initial NEDD4 studies.

This work was funded by grants from the National Institutes of Health (NIH) (NIGMS R01GM114564 to J.R. Buchan, NINDS R56NS128110 to J.R. Buchan and S.J. Barmada, and R01NS097542, and R01NS113943 to S.J. Barmada). Other support to S.J. Barmada came from the family of Angela Dobson and Lyndon Welch, the Alfred Taubman Medical Research Institute, the Danto Family, and Ann Arbor Active Against ALS. A. Byrd received support from the National Science Foundation Graduate Research Fellowship Program (DGE-1746060), L.J. Marmorale received support from an NIH training grant (T32GM136536) and an F31 (F31NS141379). M.M. Dykstra received support from an NIH F31 fellowship (F31NS134123-01), and S. Marcinowski received support from the Undergraduate Biology Research Program at the University of Arizona. Additional support came from P330AG072931 to the Michigan Alzheimer's Disease Research Center. Open Access funding provided by the University of Arizona.

Author contributions: Aaron Byrd: conceptualization, data curation, formal analysis, funding acquisition, investigation, methodology, validation, visualization, and writing—original draft, review, and editing. Lucas J. Marmorale: conceptualization, data curation, formal analysis, investigation, methodology, supervision, validation, visualization, and writing—review and editing. Sophia Marcinowski: data curation, formal analysis, investigation, methodology, supervision, visualization, and writing—review and editing. Megan M. Dykstra: data curation, formal analysis, funding acquisition, investigation, methodology, visualization, and writing—review and editing. Vanessa Addison: formal analysis, investigation, and methodology. Sami J. Barmada: formal analysis, funding acquisition, methodology, resources, supervision, visualization, and writing—review and editing. J. Ross Buchan: conceptualization, funding acquisition, methodology, project administration, resources, supervision, visualization, and writing—original draft, review, and editing.

Disclosures: All authors have completed and submitted the ICMJE Form for Disclosure of Potential Conflicts of Interest. S. Marcinowski reported "I have included some of the work present in this paper within my Master's Thesis, which is a published document though the University of Arizona. The title is "Defining Endogenous Clearance Mechanisms of Full-Length TDP-43 and Its Disease-Prone Isoforms." Link: https://search.proquest.com/openview/1c9c8927595ab4f8f164df9b01fb0f38/1?pq-origsite=gscholar&cbl=18750&diss=y." No other disclosures were reported.

Submitted: 13 December 2022

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

# Supplemental material

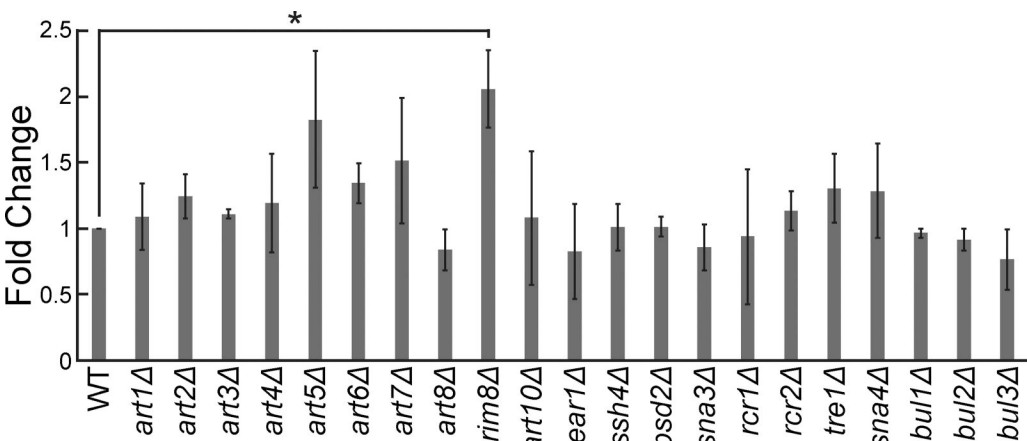

Figure S1. **Knockout of the Rsp5 adaptor Rim8 increases TDP-43-YFP levels.** Indicated strains were transformed with TDP-43-YFP plasmid (pRB109) and grown to mid-logarithmic growth phase. TDP-43-YFP levels were detected by western blot using anti–TDP-43 antibody and normalized to GAPDH loading control; quantification is shown here. *, P < 0.05; by one-way analysis of variance with Tukey's post hoc test; N = 3, except for strains *art2-4Δ, art8Δ, art10Δ, bsd2Δ, sna3Δ,* and *bul1-2Δ* (N = 2). Error bars represent standard error of the mean.

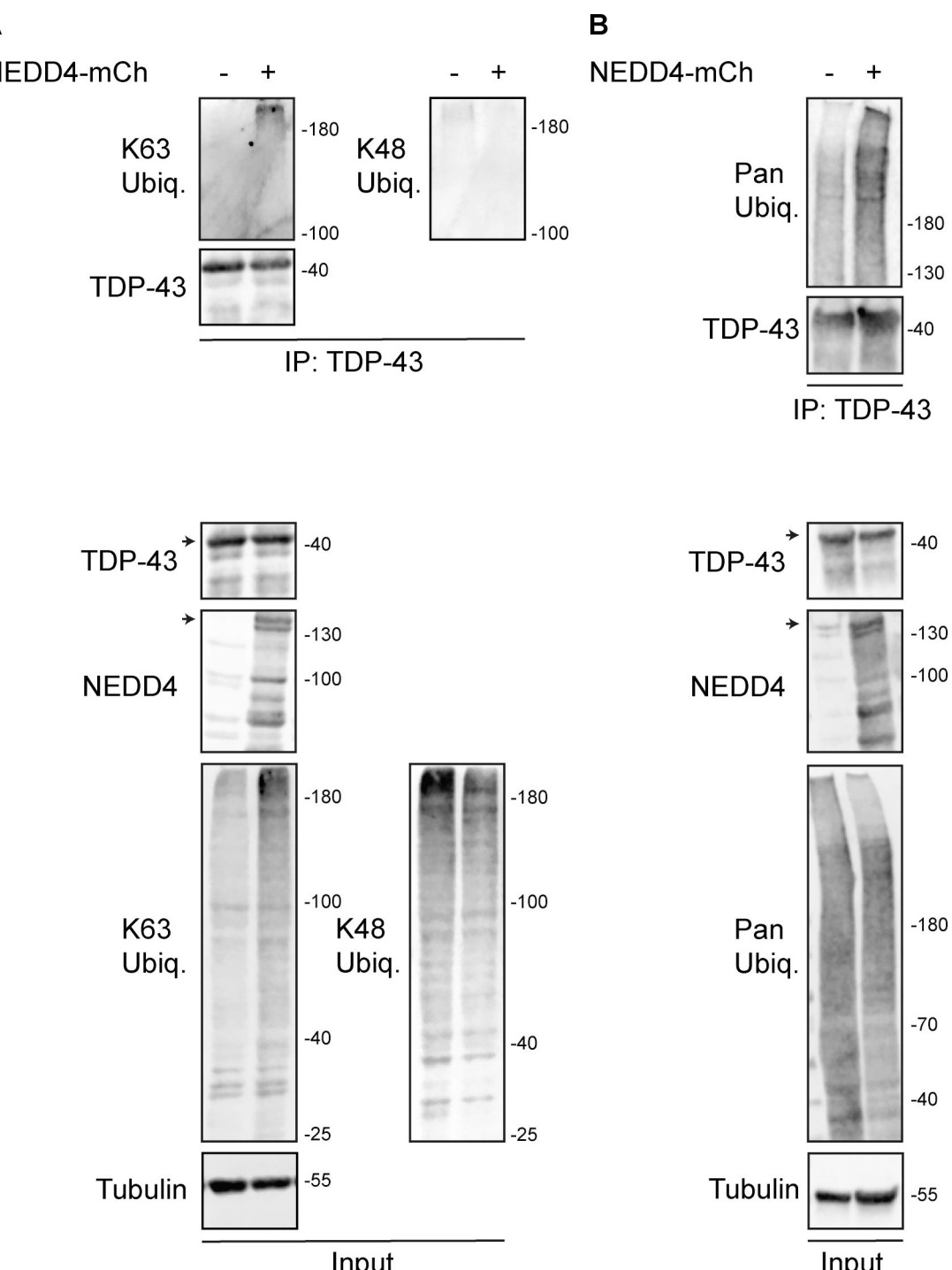

Figure S2. **HEK293A cells were transfected with HA-ubiquitin (pRB506) or both HA-Ub and NEDD4-mCherry (pRB334) and grown for 2 days before TDP-43 was immunoprecipitated. (A)** Representative biological replicate of K63 and K48 Ub signal, NEDD4, TDP-43, and α-tubulin from a single biological replicate; run on a 7% SDS-PAGE gel. **(B)** As in A, except Pan-Ub signal is shown in place of K63 and K48 on a gradient gel. Arrowheads indicate full-length species of TDP-43 and NEDD4-mCherry, with presumed degradation products/protein isoforms shown below. Trends shown were reproduced across five biological replicates. Source data are available for this figure: SourceData FS2.

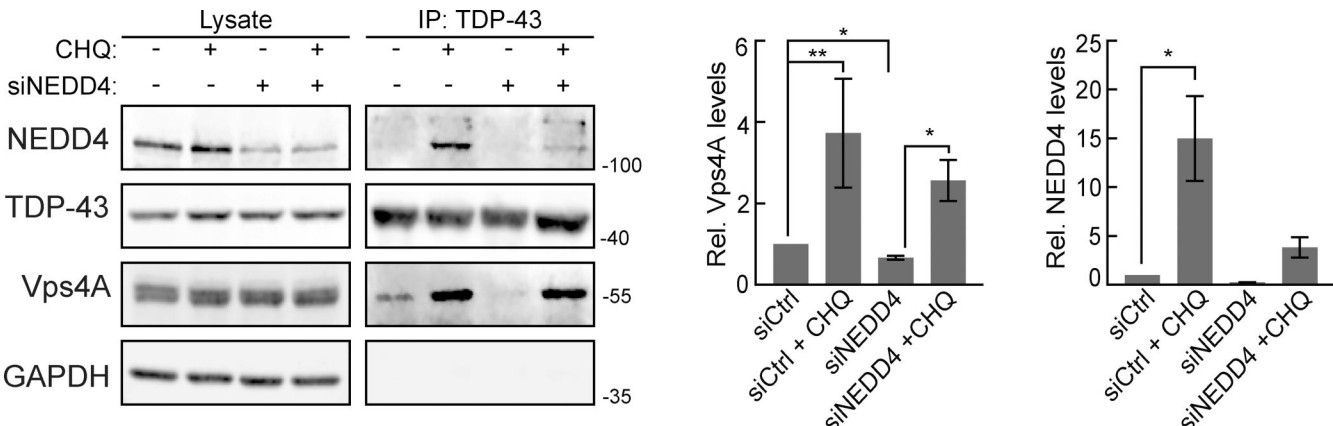

Figure S3. **Endogenous TDP-43 and VPS4A interact.** TDP-43 was immunoprecipitated in HEK293A cells in the presence or absence of CHQ treatment and/or NEDD4 knockdown, and interactions with VPS4A and NEDD4 were probed. Quantification on the right indicates VPS4A and NEDD4 pulldown under different conditions. *, P < 0.05; **, P < 0.01 by Welch's two-tailed *T* test. Error bars represent standard error of the mean. Source data are available for this figure: SourceData FS3.

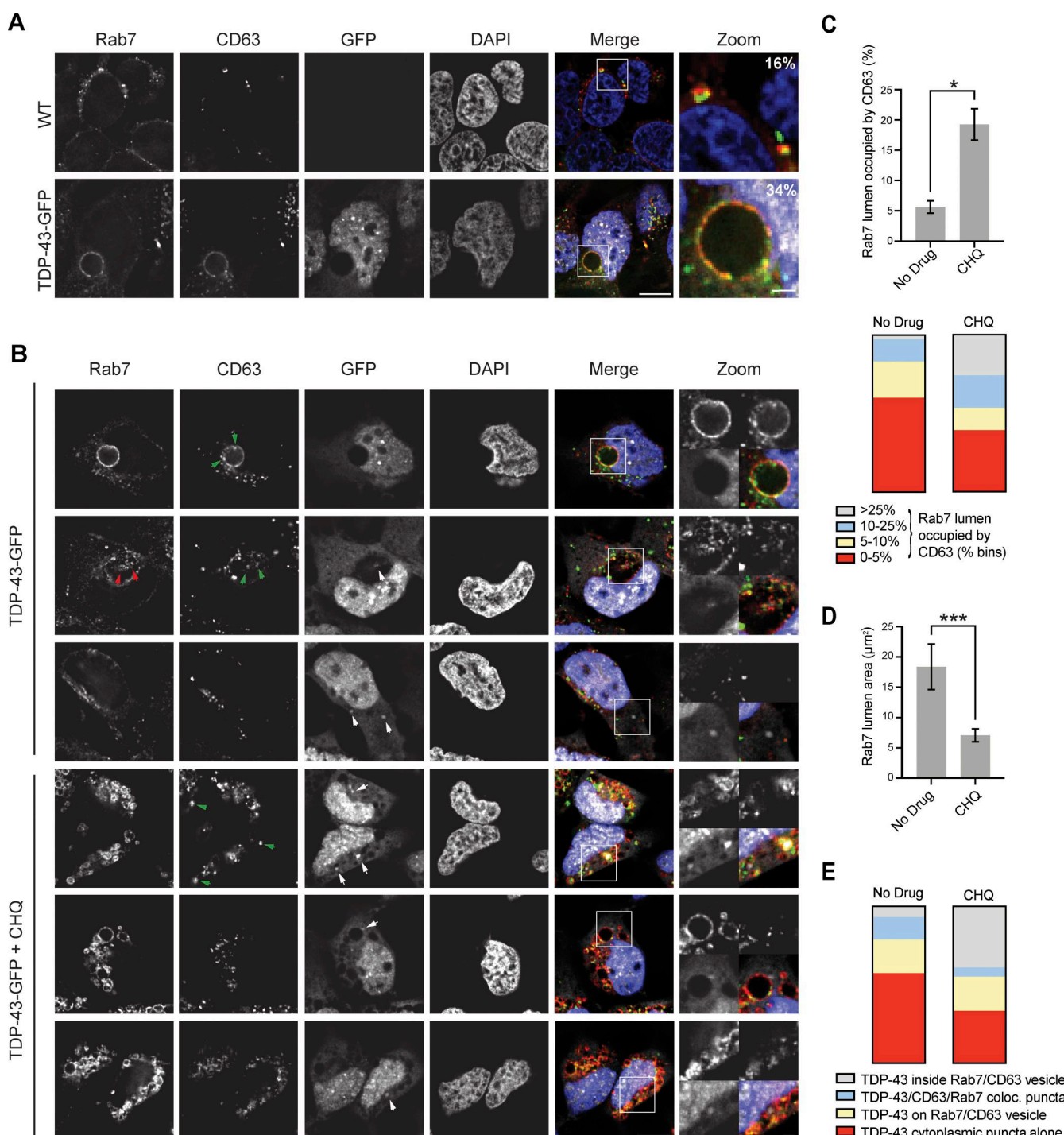

Figure S4. **Giant Rab7 vesicular structures exhibit CD63 colocalization and CHQ-sensitive intraluminal CD63 bodies, suggesting an endolysosomal origin. (A)** HEK293A cells expressing endogenous TDP-43 (WT) or TDP-43-GFP (pseudocolored white in merge) were immunostained for Rab7 (pseudocolored red) and CD63 (pseudocolored green). Percentages in zoom panels indicate Rab7 foci that overlap with CD63 foci. **(B)** TDP-43-GFP (pseudocolored white) expressing cells were immunostained for Rab7 (pseudocolored red in merge) and CD63 (pseudocolored green in merge) in the presence or absence of CHQ. Inclusion of multiple rows reflects diverse phenotypes observed. Green arrowheads indicate CD63 foci on the intraluminal Rab7 vesicular membrane (top row) or fully within Rab7 vesicles (second and fourth row). Red arrowheads indicate Rab7 puncta fully within Rab 7 vesicles (second row). White arrowheads indicate TDP-43 puncta within Rab7 vesicles (second and fourth row; bottom right in latter case), on the external Rab7 vesicular membrane (fifth row), or in distinct cytoplasmic puncta (fourth and sixth row). Scale bar, 10 μm. **(C)** Quantification of CD63 signal (average and by bins) within the lumen of Rab7 vesicles. *, P < 0.05 by Mann–Whitney U test. **(D)** Average Rab7 vesicle lumen area ± CHQ. ***, P < 0.001 by Mann–Whitney U test. **(E)** Percentage of TDP-43 localization phenotypes relative to Rab7 and CD63, binned by category. Error bars in all graphical panels represent standard error of the mean.

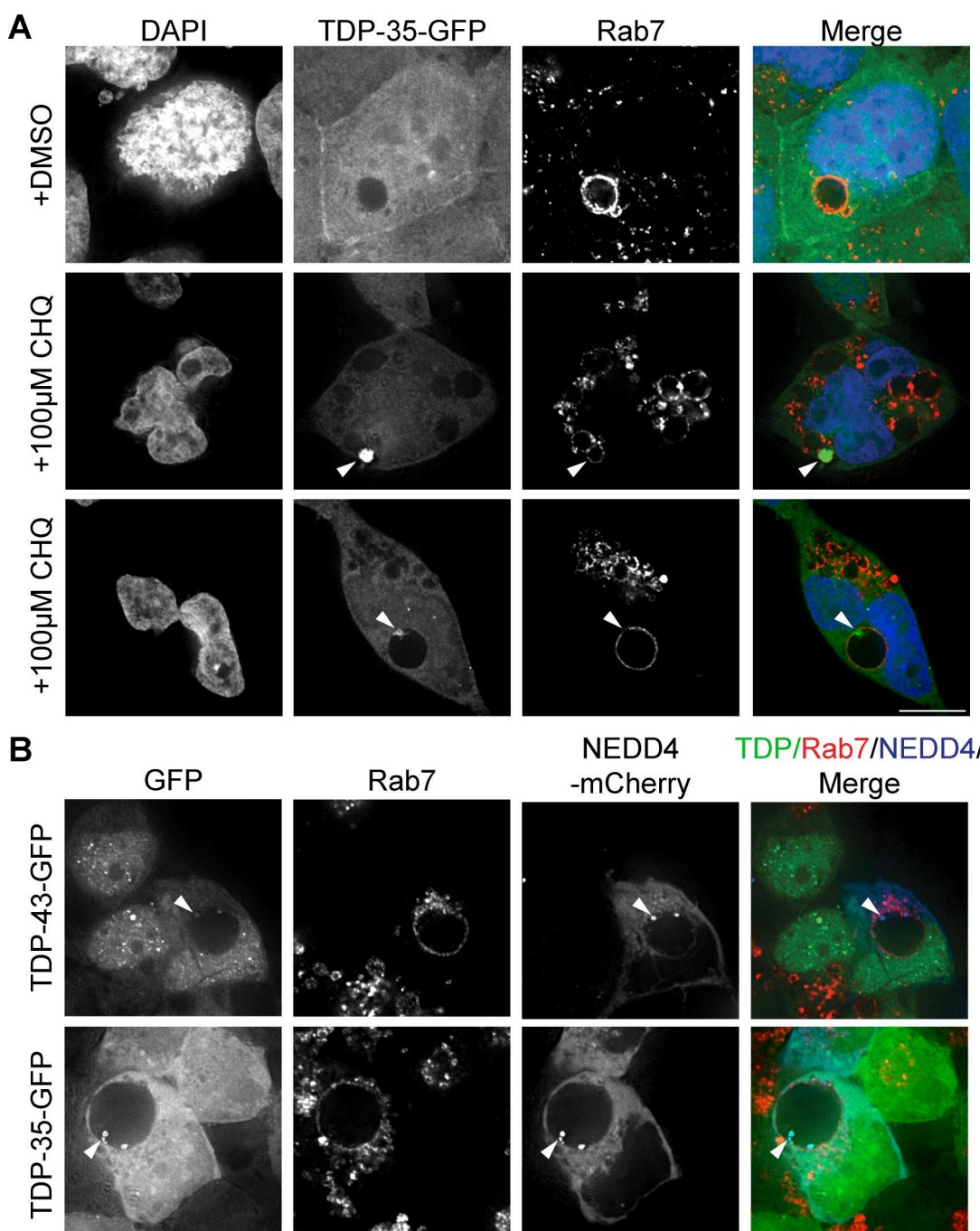

Figure S5.   **Giant Rab7-positive vesicles are proteolytically active compartments. (A)** HEK-TDP-35-GFP cells were grown in media containing solvent control (DMSO; top row) or 100 µM CHQ for 18 h (next 2 rows, to show phenotypic diversity) before being fixed and immunostained for Rab7. Arrowheads indicate TDP-35 within Rab7-positive vesicles. $N$ = 3; scale bar, 10 µm. **(B)** HEK-TDP-43-GFP and HEK-TDP-35-GFP cells were transfected with NEDD4-mCherry plasmid and grown in normal conditions for 24 h. Cells were then treated with 100 µM CHQ and incubated for 18 h before being fixed and immunostained for Rab7. Arrowheads indicate TDP-43 and TDP-35-GFP signals colocalized with NEDD4-mCherry (pseudocolored blue) within Rab7-positive giant vesicles. $N$ = 3; scale bar, 10 µm.

Provided online are Table S1, Table S2, and Table S3. Table S1 provides raw dot-blot screen data. Table S2 summarizes all quantitative data and statistical tests within the study. Table S3 summarizes yeast strains, human cell lines, plasmids, antibodies, and siRNAs used in the study.

