## [Peer Review File · The Journal of Cell Biology]

Rsp5/NEDD4 and ESCRT regulate TDP-43 toxicity and turnover via an endolysosomal clearance mechanism

Aaron Byrd, Lucas Marmorale, Sophia Marcinowski, Megan Dykstra, Vanessa Addison, Sami Barmada, and J. Buchan

Corresponding Author(s): J. Buchan, University of Arizona

Review Timeline:

Submission Date:	2022-12-13
Editorial Decision:	2023-02-14
Revision Received:	2025-06-12
Editorial Decision:	2025-07-22
Revision Received:	2025-09-15
Editorial Decision:	2025-10-03
Revision Received:	2025-10-15

Monitoring Editor: Harald Stenmark

Scientific Editor: Tim Spencer

Transaction Report:

DOI: <https://doi.org/10.1083/jcb.202212064>

February 14, 2023

Re: JCB manuscript #202212064

Dr. J. Ross Buchan
University of Arizona
Molecular and Cellular Biology
1007 E Lowell St
Tucson, Arizona 85716

Dear Dr. Buchan,

Thank you for submitting your manuscript entitled "Rsp5/NEDD4 and ESCRT regulate TDP-43 toxicity and turnover via an endolysosomal clearance mechanism". We regret the very long delays that this manuscript faced in the review process and we thank you for your patience. Your manuscript has been assessed by expert reviewers, whose comments are appended below. Although the reviewers express potential interest in this work, significant concerns unfortunately preclude publication of the current version of the manuscript in JCB.

As you will see, Reviewers praised the interesting and novel observations made in this work but all felt that evidence must be strengthened to better support multiple aspects of the authors' model. Among the numerous requests made by reviewers, three major areas stood out that must be addressed in a revision: validation that TDP-43 turnover is regulated through the proposed mechanism in neurons, stronger evidence for the ubiquitination of TDP-43, and confirmation that the Rab7 bodies observed are indeed multivesicular bodies (if possible, via EM). We also encourage the resolution of all other points made by Reviewer 1, and some kind of confirmation for this proposed mechanism using an endogenously expressed protein (related to Reviewer 2 Point 4). While all reviewer requests should be addressed in some form, additional experimental data beyond those mentioned here are not required in a revision.

Please let us know if you are able to address the major issues outlined above and wish to submit a revised manuscript to JCB. Note that a substantial amount of additional experimental data likely would be needed to satisfactorily address the concerns of the reviewers. The typical timeframe for revisions is three to four months. While most universities and institutes have reopened labs and allowed researchers to begin working at nearly pre-pandemic levels, we at JCB realize that the lingering effects of the COVID-19 pandemic may still be impacting some aspects of your work, including the acquisition of equipment and reagents. Therefore, if you anticipate any difficulties in meeting this aforementioned revision time limit, please contact us and we can work with you to find an appropriate time frame for resubmission. Please note that papers are generally considered through only one revision cycle, so any revised manuscript will likely be either accepted or rejected.

If you choose to revise and resubmit your manuscript, please also attend to the following editorial points. Please direct any editorial questions to the journal office.

GENERAL GUIDELINES:

Text limits: Character count is < 40,000, not including spaces. Count includes title page, abstract, introduction, results, discussion, and acknowledgments. Count does not include materials and methods, figure legends, references, tables, or supplemental legends.

Figures: Your manuscript may have up to 10 main text figures. To avoid delays in production, figures must be prepared according to the policies outlined in our Instructions to Authors, under Data Presentation, <https://jcb.rupress.org/site/misc/ifora.xhtml>. All figures in accepted manuscripts will be screened prior to publication.

IMPORTANT: It is JCB policy that if requested, original data images must be made available. Failure to provide original images upon request will result in unavoidable delays in publication. Please ensure that you have access to all original microscopy and blot data images before submitting your revision.

Supplemental information: There are strict limits on the allowable amount of supplemental data. Your manuscript may have up to 5 supplemental figures. Up to 10 supplemental videos or flash animations are allowed. A summary of all supplemental material should appear at the end of the Materials and methods section.

Please note that JCB now requires authors to submit Source Data used to generate figures containing gels and Western blots with all revised manuscripts. This Source Data consists of fully uncropped and unprocessed images for each gel/blot displayed in the main and supplemental figures. Since your paper includes cropped gel and/or blot images, please be sure to provide one Source Data file for each figure that contains gels and/or blots along with your revised manuscript files. File names for Source Data figures should be alphanumeric without any spaces or special characters (i.e., SourceDataF#, where F# refers to the

associated main figure number or SourceDataFS# for those associated with Supplementary figures). The lanes of the gels/blots should be labeled as they are in the associated figure, the place where cropping was applied should be marked (with a box), and molecular weight/size standards should be labeled wherever possible.

If you choose to resubmit, please include a cover letter addressing the reviewers' comments point by point. Please also highlight all changes in the text of the manuscript.

Regardless of how you choose to proceed, we hope that the comments below will prove constructive as your work progresses. We would be happy to discuss them further once you've had a chance to consider the points raised. You can contact the journal office with any questions, cellbio@rockefeller.edu or call (212) 327-8588.

Thank you for thinking of JCB as an appropriate place to publish your work.

Sincerely,

Harald Stenmark
Monitoring Editor
Journal of Cell Biology

Tim Fessenden
Scientific Editor
Journal of Cell Biology

Reviewer #1 (Comments to the Authors (Required)):

In this manuscript, Byrd et al. investigate the roles of endosomal pathway in regulating TDP-43 clearance. Using unbiased yeast genetic screening, the authors identify the genes involved in regulating TDP-43 stability. Gene ontology analyses reveal that K63-linked ubiquitination and endosome are the top pathways for TDP-43 clearance. Moreover, the authors show that modulations of the E3 ligase Rsp5/NEDD4 and core ESCRT protein VPS4 in yeast and cultured cells influence TDP-43 turnover, mislocalization and toxicity. Furthermore, the authors show that NEDD4 binds to TDP-43 and mediates TDP-43 ubiquitination. Lastly, the authors show that overexpression of TDP-43 results in formation of giant MVBs. Interestingly, TDP-43 accumulates in these giant MVBs in a NEDD4-dependent manner. This manuscript is interesting and novel. The authors provide a novel mechanism of how endolysosome regulates TDP-43 clearance. However, the present data in its current format is insufficient to support the conclusions. The specific points are listed below.

1. The main findings about NEDD4 and VPS4 mediating TDP-43 turnover, mislocalization and toxicity need to be validated in human neuron. Moreover, the colocalization of TDP-43 and NEDD4, or VPS4 also needs to be validated in human neurons.
2. This manuscript contains two parts. The first part shows endosome pathway mediates TDP-43 turnover, mislocalization and toxicity. The second part shows that overexpression of TDP-43 causes MVBs abnormalities. However, the molecular connection of these two parts is unclear. Better interaction is warranted.
3. In Figure 2E, it seems that TDP-43 levels are higher in cells expressing GFP-VPS28 or GFP-VPS36 compared to WT in input. Is this result expected?
4. In Figure 3E, it seems that TDP-43 levels are higher in cells expressing GFP-Rsp5 compared to WT in input. Is this result expected? Lastly, in Figure 3G, the authors need to perform quantitative analysis of TDP-43 levels in cells expressing Rsp5. Based on the model proposed in Figure 7, the authors need to determine whether knockdown/knockout VPS4 influence Rsp5-mediated TDP-43 clearance.
5. In Figure 4B, the authors need to include Rab7 staining. This result will bridge the gap discussed in the comment #2.
6. In Figure 5F, the authors need to perform ubiquitination assay using ubiquitin mutant to confirm NEDD4-mediated TDP-43 ubiquitination is K63-linked.
7. In Figure 6, it is unclear why TDP-43 overexpression affect MVBs. Moreover, does overexpression of also affect early endosome or lysosome? Lastly, since Rab7 also expresses in late endosome and lysosome, so the authors need more specific MVB markers to validate their Rab7 staining.
8. In Figure 7, the author needs to perform EM to examine the structure of giant MVBs and immuno-EM to examine whether

TDP-43 accumulates in giant MVBs. Moreover, in Figure 7B, the authors show that knockdown NEDD4 reduces TDP-43 localization in giant MVBs. The authors need to examine whether overexpression of NEDD4 increases intraluminal TDP-43 in giant MVBs.

9. In Figure 8, the authors posit that the ubiquitination of TDP-43 recruits the ESCRT complex, which ultimately mediates its sorting to MVBs. But the claim is not supported experimentally. An experiment in which NEDD4 knock-down abrogates the interaction of TDP-43 with ESCRT factors would provide experimental validation that ubiquitination is required to start the endosomal sorting process. Moreover, the authors show that VPS28 and VPS36 binds to non-ubiquitinated TDP-43 (Figure 2E), which also challenges the proposed model.

Reviewer #2 (Comments to the Authors (Required)):

The manuscript entitled "Rsp5/NEDD4 and ESCRT regulate TDP-43 toxicity and turnover via an endolysosomal clearance mechanism" is very interesting and the use of yeast to screen for modifiers of TDP-43 turnover is an excellent approach to identify conserved pathways that impact levels of TDP-43 levels. The focus on the top ten hits in the ESCRT pathway is well justified and the validation in yeast of the Rsp5/NEDD4 and ESCRT is generally reasonable. The main weakness of this manuscript is many of the experiments are done with GFP-fused proteins and not with endogenously expressed proteins. Further the pathways evaluated are not looked at in neurons which is incredibly important to validate. It maybe that these findings are not relevant to turnover of TDP-43 at endogenous levels and some of findings are artificial. Points to address in manuscript:

- (1) Need reference(s) for this statement. "In yeast and human cell models, impairment of endolysosomal flux caused increased TDP-43 accumulation, aggregation, and toxicity, whereas enhancing endolysosomal flux suppressed these phenotypes."
- (2) The screen carefully describes the controls in the text but not the yeast gene deletion library. Maybe standard in the yeast field but some general information needs to be provided. This is relevant to how many hits. So how many total strains as well.
- (3) Validation in a neuronal model expressing endogenous levels of TDP-43 modulating Rsp5/NEDD4 and ESCRT would strengthen the paper.
- (4) Protein interactions described in this paper should be done with endogenous levels of the proteins and the interactions determined with pull-down of each putative partner. Confocal to validate this interaction and quantification.

Reviewer #3 (Comments to the Authors (Required)):

Byrd et al. examine the clearance of cytoplasmic TDP43 in cells using both yeast and HEK293-based models. Yeast genetic screens reveal that ESCRT genes and Rsp5/NEDD4 are required for efficient TDP43 clearance. Analogous roles for the ESCRT pathway, particularly mammalian VPS4 and NEDD4 in the turnover of TDP43 are proposed in the HEK293 model. The authors also find that overexpression of GFP-tagged TDP43 or a disease-associated isoform, TDP-35, leads to a profound increase in the size of Rab7 positive vesicles, which they propose to represent an impairment in late endosomes/MVBs trafficking to lysosomes. These vesicles accumulate TDP43-GFP, but only upon treatment with the lysosome inhibitor, HCQ, suggesting that TDP43-GFP is being degraded (or the green fluorescence quenched) in this compartment. However, upon NEDD4 knockdown, TDP43-GFP does not accumulate in HCQ-treated Rab7 vesicles. Based on these results, the authors propose that NEDD4-mediated ubiquitination of TDP43 results in its ESCRT-dependent incorporation into intraluminal vesicles, resulting in its endolysosomal clearance in process that resemble endosomal micro-autophagy (eMI). Although the genetics support a role for ESCRTs and NEDD4 in cytoplasmic TDP43 clearance, the cell biological studies are low resolution in scope. The author's model requires additional experimental support that TDP43 is being incorporated into intraluminal vesicles (ILVs).

- 1) Although Rab7 localizes to late endosomes/MVBs, it is not uniquely present at this compartment in cells. Thus, additional evidence is needed to support that the large vesicles in TDP43 overexpressing cells are late endosomal in origin. Additional LE/MVB markers such as CD63 and and LBPA should be analyzed in Figures 5 and 6.
- 2) Transmission electron microscopy of the Rab7 positive vesicles in TDP43- or TDP35-overexpressing cells should be conducted to confirm that these organelles exhibit evidence of ILVs as proposed by the authors in their model. Furthermore, immuno-EM using GFP antibodies will help corroborate that TDP-43 is located at these ILVs These are critical experiments to corroborate that the large vacuoles induced by TDP43 are MVBs that contain ILVs with TDP-43 as cargo.
- 3) Similarly, for the studies of NEDD4 siRNA in Fig 5A, the effects of HCQ treatment on TDP-43 protein turnover following cycloheximide treatment should be analyzed via immunoblotting to confirm the role of endolysosomal acidification on the TDP43 and TDP35 clearance.
- 4) Immunofluorescence analysis for K63-ubiquitin chains within these large vesicle should be performed in Fig 7B-C to potentially support the author's model.
- 5) Does NEDD4 and VPS4 control TDP43 secretion into the conditioned media via EVs? In eMI, substrates are not only degraded but secreted outside of a cell via extracellular vesicles (EVs).
- 6) The analysis of "cell viability" in Fig 3D and 4C is based on enumerating cell numbers. However, this does not distinguish between reduced cell proliferation versus increased cell death in these cultures. More precise measures of cell death, either

measurement of apoptotic markers or dye exclusion in cells, should be analyzed to draw conclusions regarding cell viability or survival in these assays.

Minor issues:

- 1) The cycloheximide chase immunoblots in Fig 3A are not particularly convincing for turnover for TDP-43 the controls. Longer chase periods (72h) should be considered.
- 2) The immunoblot in Fig 4B is not particularly convincing for a reduction in TDP43 following NEDD4 overexpression. A cycloheximide chase should be performed similar to the NEDD4 knockdown.

Reviewer comments and our responses

We sincerely thank the reviewers for their very helpful comments and apologize for the long delay in addressing them. This reflected various factors (1st author moved 2 weeks after feedback received, a new student took over the work, optimizing new experimental approaches, personal circumstances etc.). Regardless, we are confident the manuscript has been improved as a result of your feedback and thank you for your efforts.

Reviewer #1:

In this manuscript, Byrd et al. investigate the roles of endosomal pathway in regulating TDP-43 clearance. Using unbiased yeast genetic screening, the authors identify the genes involved in regulating TDP-43 stability. Gene ontology analyses reveal that K63-linked ubiquitination and endosome are the top pathways for TDP-43 clearance. Moreover, the authors show that modulations of the E3 ligase Rsp5/NEDD4 and core ESCRT protein VPS4 in yeast and cultured cells influence TDP-43 turnover, mislocalization and toxicity. Furthermore, the authors show that NEDD4 binds to TDP-43 and mediates TDP-43 ubiquitination. Lastly, the authors show that overexpression of TDP-43 results in formation of giant MVBs. Interestingly, TDP-43 accumulates in these giant MVBs in a NEDD4-dependent manner. This manuscript is interesting and novel. The authors provide a novel mechanism of how endolysosome regulates TDP-43 clearance. However, the present data in its current format is insufficient to support the conclusions. The specific points are listed below.

Response: We thank the reviewer for stating our manuscript is interesting and novel, and that we present a novel mechanism for TDP-43 clearance.

1. The main findings about NEDD4 and VPS4 mediating TDP-43 turnover, mislocalization and toxicity need to be validated in human neuron. Moreover, the colocalization of TDP-43 and NEDD4, or VPS4 also needs to be validated in human neurons.

Response: These are good ideas, and similar to a request from reviewer 2 (point 3), who specifically asked that we look at impacts of NEDD4 and ESCRT on endogenous TDP-43. We thus collaborated with Sami Barmada's lab (University of Michigan) and examined the turnover in their iNeuron model (e.g., Weskamp et al, JCI 2020) of endogenously expressed TDP-43 C-terminally fused to Dendra2, a photoconvertible protein that enables in vivo measurement of protein stability through optical pulse labelling. This approach avoids potential off-target effects of cycloheximide pulse-chase assays, where global protein synthesis is inhibited. Though NEDD4 knockdown via virally transduced shRNA was somewhat inefficient (~65%), we still observed a significant stabilization of TDP-43 in the optical pulse chase assay (New Figure 6). Note, this stabilization may even be an underestimate not just because of incomplete NEDD4 KD, but also because the optical pulse labeling assay described here does not discriminate between nuclear and cytoplasmic TDP-43. While it is true that TDP-43 undergoes nucleocytoplasmic shuttling, the large majority of the protein remains in the nucleus. Our data suggest that NEDD4 primarily affects cytoplasmic TDP-43; therefore, the observed stabilization of TDP-43 in these experiments may reflect effects on a small proportion of the total pool of cellular TDP-43. Compartment-

specific analysis of TDP-43 turnover is something that Barmada and Buchan labs plan to look at in future work together.

We appreciate the reviewers other suggestions, including examining VPS4 effects in neurons. We focused solely on NEDD4 for several reasons.

- 1. In our view, there is greater novelty in studying NEDD4 in the context of TDP-43 turnover and ALS progression (and possibly later therapeutic exploration). ESCRT factors, MVBs and Vps4 have been linked both to TDP-43 turnover (Filimonenko et al, JCB 2007) and ALS pathogenesis via nuclear pore complex injury, (Coyne et al, Rothstein, Sci Trans Med, 2021; Coyne & Rothstein, Acta Neuropathologica Communications, 2021; Dubey et al, Lloyd, Cell Reports, 2022), thus potentially complicating interpretations.*
- 2. We hypothesize that NEDD4's effect is more proximal and specific to TDP-43 than VPS4/ESCRT function, given that an E3 ligase could directly modify TDP-43, and that ESCRT proteins generally bind and facilitate internalization/remodeling of ubiquitinated proteins to or from membranous organelles.*
- 3. Vps4 has two paralogs in human cells that are likely to be at least partially functionally redundant (Szymanska et al, Miaczynska, EMBO Mol Med, 2020; Neggers et al, Aguirre, Cell Reports 2020), thus complicating knockdown experiments.*

The reviewer makes valuable suggestions regarding examining NEDD4 and VPS4 impacts on TDP-43 localization, toxicity and potential co-localization of TDP-43 with each factor. Regarding co-localization, TDP-43 cytoplasmic aggregate formation is very limited in unperturbed iNeurons at endogenous expression levels, making co-localization experiments largely infeasible. The other points are areas of future interest, which we respectfully suggest are beyond the scope of the current study, given the volume of data in the manuscript in two other model systems.

2. This manuscript contains two parts. The first part shows endosome pathway mediates TDP-43 turnover, mislocalization and toxicity. The second part shows that overexpression of TDP-43 causes MVBs abnormalities. However, the molecular connection of these two parts is unclear. Better interaction is warranted.

Response: See response to point 5 below, as the reviewer indicated conducting the experiment outlined would address this point.

Additionally, as noted in our discussion, and based in part on our work from 2017 (Liu et al, Nat Comm), we suspect that TDP-43 clearance by endolysosomal means can be at a certain threshold overwhelmed, perhaps by sequestration of endolysosomal trafficking factors (like ESCRT and Rab proteins) into TDP-43 cytoplasmic aggregates. We see increasing defects in endocytic flux in response to expression of increasing aggregate prone forms of TDP-43 (Liu et al, Buchan, 2017, Fig 5a) and accumulation of endolysosomal factors in TDP-43 aggregates (Fig 5b-c of same paper; also yeast data in that manuscript, and our current work). Thus, a simple working model is that as TDP-43 clearance by endolysosomal means fails, subsequent sequestration of endolysosomal factors could lead to endolysosomal trafficking

defects such as the giant Rab7-positive vesicles we observe. Further testing of this and other models is an area of future interest.

3. In Figure 2E, it seems that TDP-43 levels are higher in cells expressing GFP-VPS28 or GFP-VPS36 compared to WT in input. Is this result expected?

Response: We thank the reviewer for noticing this. We do not have an obvious explanation for this result. It is possible that GFP tagging of Vps28 and Vps36 impairs ESCRT function, and thus TDP-43 turnover, hence the increase in steady state TDP-43 levels. Additionally, reviewer 2 had concerns (point 4) about IP data in which interacting proteins were not expressed at endogenous/physiological levels; the GFP-tagged ESCRT proteins are driven in this experiment by a non-native promoter (NOP1) that may induce their over-expression. We also cannot tell from this assay whether TDP-43-Vps protein interactions are functional events in TDP-43 turnover, or due to TDP-43 sequestering Vps proteins in aggregates (as potentially suggested by Figure 2D); we have now noted this caveat in the results text (p8, end of 1st paragraph). Thus, for all these reasons, we decided to remove this data. Note, as suggested by reviewer 2, point 4, we have obtained endogenous TDP-43 and Vps4A IP interaction data in human cells (New Supplementary Figure 3), thus improving on our original yeast data.

4. In Figure 3E, it seems that TDP-43 levels are higher in cells expressing GFP-Rsp5 compared to WT in input. Is this result expected? Lastly, in Figure 3G, the authors need to perform quantitative analysis of TDP-43 levels in cells expressing Rsp5. Based on the model proposed in Figure 7, the authors need to determine whether knockdown/knockout VPS4 influence Rsp5-mediated TDP-43 clearance.

Response: We thank the reviewer for noticing this. For similar reasons as outlined above with our old Figure 2E, we decided to remove this data.

Regarding our old Figure 3G, we did previously quantify the impact on TDP-43 levels of Rsp5 OE (or normal expression level). However, this data has now been replaced and quantified (see next paragraph).

Checking the impact of Rsp5 OE in a Vps4 KO background is an excellent idea; thus, we transformed our Rsp5 OE plasmid into Vps4 KO cells and Vps28 KO cells (ESCRT-I subunit) and determined if the ability of Rsp5 to reduce TDP-43 levels as in WT was affected. As predicted by our model, the reduction in TDP-43 levels caused by Rsp5 overexpression is attenuated by the absence of either Vps4 or Vps28. We have replaced the original Figure 3G panel with this new data (Figure 3F) and have added quantitation of the three biological replicates.

5. In Figure 4B, the authors need to include Rab7 staining. This result will bridge the gap discussed in the comment #2.

Response: This is a good idea, which we have addressed – however, because of the complexity of our data, we are hesitant to include in the paper at present since we lack a clear understanding of what it means.

6. In Figure 5F, the authors need to perform ubiquitination assay using ubiquitin mutant to confirm NEDD4-mediated TDP-43 ubiquitination is K63-linked.

Response: This is a good point. Initially, we tried using a collection of HA-tagged Ub overexpression plasmids from the Dawson lab (Lim et al, J Neurosci 2005) that contained either WT Ub, or mutant Ub versions that only allowed on K63, K48 or no (K0) polyubiquitin chains to form following their incorporation. Though results were suggestive of a NEDD4-dependent K63 bias, as has been reported elsewhere (see additional text in results section, p12, 3rd paragraph), we struggled with significant variability in transfection efficiency and/or Ub expression level (assessed by anti-HA blotting) between the different Ub plasmids.

Thus, we switched focus to repeating our TDP-43 IP analyses in Figure 5, but with K63 and K48-specific antibodies, as well as a Pan-Ub antibody. Owing to difficulties we were experiencing at the time with NEDD4 knockdown efficiency, we decided to simply focus on WT cells and those over-expressing NEDD4. In addition, given general concerns amongst reviewers about possible TDP-43 overexpression artefacts, we switched to examining endogenous TDP-43 ubiquitination. Across 5 biological replicates (new Supplemental Figure 2), we consistently observed that NEDD4 overexpression increased TDP-43 K63 and pan-Ubiquitin levels (the latter result mirroring our findings in Figure 5F). K48 Ubiquitination signal on TDP-43 was very weak, and actually decreased following NEDD4 over-expression. This result suggests NEDD4 at least partly ubiquitinates TDP-43 via K63-linked polyubiquitin chains.

7. In Figure 6, it is unclear why TDP-43 overexpression affect MVBs. Moreover, does overexpression of TDP-43 also affect early endosome or lysosome? Lastly, since Rab7 also expressed in late endosome and lysosome, so the authors need more specific MVB markers to validate their Rab7 staining.

Response: We do not know precisely why TDP-43 overexpression affects MVBs. However, we do speculate on this in our discussion (p20; 2nd paragraph). In short, our prior work has demonstrated endocytosis rate defects associated with our TDP-43-GFP HEK293 cell models, particularly lines which express C-terminal fragments of TDP-43 that are more aggregation prone and likely to accumulate in the cytoplasm (Liu et al, Nat Com, 2017). Work from Wilfred Rossol's lab using proximity labelling of

aggregate-prone TDP-43 also shows a striking enrichment of vesicular trafficking proteins within TDP-43 aggregates; we have seen this too for select candidates via microscopy (Liu et al, Nat Com, 2017; and this work). Thus, the simplest model in our view is that TDP-43 aggregates can sequester endolysosomal factors, including potentially those that regulate MVB remodeling events.

Regarding potential impacts of modest TDP-43 overexpression on early endosomes and lysosomes, we have only currently examined Rab5 localization in WT and TDP-43 over-expressing cells. We do not see striking changes in Rab5 foci size or localization. Given this, we prefer to focus on the very obviously distinct Rab7 phenotype in our paper for now.

To the final point, as also raised by reviewer 3 (point 1), we added a new panel C to Supplementary Figure 4 showing that the giant vesicles were MVB-like in nature by co-staining Rab7 with CD63, another MVB marker. In cells with both endogenous TDP-43 expression levels, and modest overexpression (TDP-43-GFP expressing cells), co-localization between Rab7 and CD63 is observed, particularly with the giant Rab7-positive vesicles, much more so than what we see with Rab7 and other endolysosomal markers in Figure 7B (i.e., Rab5 (early endosome) LAMP1 (lysosome) and LC3 (autophagosome)). This data is described on p20m second paragraph, and supports the notion that these vesicles are MVB-like based on composition. However, morphological differences clearly indicate these are not conventional MVBs (a point we also highlight in the results text).

8. In Figure 7, the author needs to perform EM to examine the structure of giant MVBs and immuno-EM to examine whether TDP-43 accumulates in giant MVBs. Moreover, in Figure 7B, the authors show that knockdown NEDD4 reduces TDP-43 localization in giant MVBs. The authors need to examine whether overexpression of NEDD4 increases intraluminal TDP-43 in giant MVBs.

Response: EM is a good suggestion (also raised by Reviewer 3, point 2), and one which we tried, repeatedly, including immunolabelling. Regrettably, our EM core facilities proved inadequate in their support and ability to help us obtain images of sufficient quality to make reliable interpretations with. We would like to pursue this at an external facility moving forward in future work, but costs are currently prohibitive to us. Given that we did additional IF analyses with CD63 to confirm that our giant Rab7 positive structures are MVB-like in their composition (point 7 above), and that our IF data already indicates TDP-43 can be localized within Rab7-positive vesicles, we respectfully suggest that EM is not critical at this time.

To the point regarding whether NEDD4 overexpression increases TDP-43 signal in giant Rab7-positive vesicles this is a useful idea, but our attempts at such quantification lead to concerns that significant variability in NEDD4 expression due to variable transfection efficiency precluded a clear conclusion. We also suspect that TDP-43 targeting to “normal” Rab7-positive MVBs may be occurring, but our microscopy data lacks resolution to quantify this unambiguously. We have added a statement indicating this in our discussion (p23, 2nd paragraph), as well as an acknowledgement that NEDD4 could in principle clear TDP-43 by other non-endolysosomal means. Nonetheless, the totality of our other data presented here (including NEDD4 overexpression, endogenous NEDD4 levels, or KD) present a reasonable argument that NEDD4’s impact on TDP-43 turnover depends at least partly via targeting to endolysosomal organelles.

9. In Figure 8, the authors posit that the ubiquitination of TDP-43 recruits the ESCRT complex, which ultimately mediates its sorting to MVBs. But the claim is not supported experimentally. An experiment

in which NEDD4 knock-down abrogates the interaction of TDP-43 with ESCRT factors would provide experimental validation that ubiquitination is required to start the endosomal sorting process. Moreover, the authors show that VPS28 and VPS36 binds to non-ubiquitinated TDP-43 (Figure 2E), which also challenges the proposed model.

Response: NEDD4 KD and examining ESCRT factor interaction with TDP-43 is indeed a good idea. Having successfully demonstrated the interaction of endogenous TDP-43 with Vps4A (and NEDD4), as per the suggestion of reviewer 2 (point 4), we also examined the effect of NEDD4 siRNA knockdown on TDP-43 and Vps4 interaction. Interestingly, NEDD4 KD reduces TDP-43's interaction with Vps4 by about 30-35%. We have added a new supplemental figure (Supplemental Figure 3) and results text section describing these results, and full quantification of TDP-43, NEDD4, and Vps4A in lysates and IPs to supplemental table 2.

To the point regarding TDP-43 binding Vps28 and 36 in a non-ubiquitinated state, as noted earlier in our response to point 3, we have removed Figure 2E data due to various concerns with interpretation.

Reviewer #2 (Comments to the Authors (Required)):

The manuscript entitled "Rsp5/NEDD4 and ESCRT regulate TDP-43 toxicity and turnover via an endolysosomal clearance mechanism" is very interesting and the use of yeast to screen for modifiers of TDP-43 turnover is an excellent approach to identify conserved pathways that impact levels of TDP-43 levels. The focus on the top ten hits in the ESCRT pathway is well justified and the validation in yeast of the Rsp5/NEDD4 and ESCRT is generally reasonable. The main weakness of this manuscript is many of the experiments are done with GFP-fused proteins and not with endogenously expressed proteins. Further the pathways evaluated are not looked at in neurons which is incredibly important to validate. It maybe that these findings are not relevant to turnover of TDP-43 at endogenous levels and some of findings are artificial. Points to address in manuscript:

Response: We thank the reviewer for their positive comments.

(1) Need reference(s) for this statement. "In yeast and human cell models, impairment of endolysosomal flux caused increased TDP-43 accumulation, aggregation, and toxicity, whereas enhancing endolysosomal flux suppressed these phenotypes."

Response: We have added references to our previous works immediately at the end of this sentence.

(2) The screen carefully describes the controls in the text but not the yeast gene deletion library. Maybe standard in the yeast field but some general information needs to be provided. This is relevant to how many hits. So how many total strains as well.

Response: This is a good point. We have added references to the yeast gene deletion library both in results text (p5) and in the methods section (p25). Out of 4944 non-essential gene deletion strains that exist in the collection, we successfully transformed and screened 3663 strains in ≥ 3 replicate dot blot screens (74% of all non-essential genes). We have also added this information to the methods section (p25).

(3) Validation in a neuronal model expressing endogenous levels of TDP-43 modulating Rsp5/NEDD4 and ESCRT would strengthen the paper.

Response: This is a good suggestion, similar to that of Reviewer 1 (point 1). As previously stated in point 1 "We thus collaborated with Sami Barmada's lab (University of Michigan) and examined the turnover in their iNeuron model (e.g. Weskamp et al, JCI 2020) of endogenously expressed TDP-43 C-terminally fused to Dendra2, a photoconvertible protein that enables in vivo measurement of protein stability through optical pulse labelling. This approach avoids potential off-target effects of cycloheximide pulse-chase assays, where global protein synthesis is inhibited. Though NEDD4 knockdown via virally transduced shRNA was somewhat inefficient (~65%), we still observed a significant stabilization of TDP-43 in the optical pulse chase assay (New Figure 6)."

Please see our more detailed response under that comment.

(4) Protein interactions described in this paper should be done with endogenous levels of the proteins and the interactions determined with pull-down of each putative partner. Confocal to validate this interaction and quantification.

Response: This is a good point and affects our yeast TDP-43-GFP interaction datasets with GFP-tagged Vps28 and 36 (old Fig 2E) and GFP-Rsp5 (old Fig 3E). TDP-43 by default has no endogenous expression level in yeast, but the Vps and Rsp5 factors are likely over-expressed given the fairly strong Nop1 promoter they are driven by. Given separate concerns from reviewer 1 about TDP-43 expression levels varying in WT cells, versus in yeast +/- GFP-Vps28, GFP-Vps36 and GFP-Rsp5 (points 3 and 4), we decided to cut this data from the paper given the reviewer-identified caveats, and because we don't feel it is that critical. It is also hard to say if these interactions are "functional", or the consequence of TDP-43 being sequestered in aggregates with Vps factors (as Figure 2D might suggest).

For the sole human Co-IP experiment in the paper between TDP-43 and NEDD4, previously this was done in HEK293A cells with overexpression of NEDD4-mCh and modest over-expression of TDP-43-GFP (Figure 5E). We have kept this in, but as suggested, we examined endogenous protein interactions between TDP-43 and ESCRT and NEDD4 in HEK293A cells.

Importantly, we identified a robust interaction between endogenous TDP-43 and VPS4, which increases in CHQ-treated cells, possibly reflective of an impairment in TDP-43 turnover by endolysosomal means (Supplemental Figure 3). Interestingly, this interaction also decreases in the presence of NEDD4 KD (also discussed in response to reviewer 1, point 9).

We were also able to detect a very weak interaction between TDP-43 and NEDD4, which strengthened significantly with CHQ treatment (Supplemental Figure 3, Supplemental table 2). We would also note that TDP-43 was recently identified as a putative NEDD4 substrate in proteomic studies, details of which we have added to our discussion (p23, 2nd paragraph).

Regarding "confocal to validate interaction", naturally most standard immunofluorescence microscopy methods cannot directly demonstrate physical interactions, just proximity. In this regard though, we have previously shown that endogenously expressed TDP-43 and exogenously expressed dominant negative Vps4 form colocalized foci. Regarding endogenous NEDD4 and TDP-43, there are very few cytoplasmic TDP-43 foci to quantify in HEK293A cells. There is some diffuse cytoplasmic signal overlap between TDP-43 and NEDD4, but the relevance of this "colocalization" is hard to gauge, thus we chose not to perform quantification of NEDD4 and TDP-43 in HEK293A cells under normal growth.

Reviewer #3:

Byrd et al. examine the clearance of cytoplasmic TDP43 in cells using both yeast and HEK293-based models. Yeast genetic screens reveal that ESCRT genes and Rsp5/NEDD4 are required for efficient TDP43 clearance. Analogous roles for the ESCRT pathway, particularly mammalian VPS4 and NEDD4 in the turnover of TDP43 are proposed in the HEK293 model. The authors also find that overexpression of GFP-tagged TDP43 or a disease-associated isoform, TDP-35, leads to a profound increase in the size of Rab7 positive vesicles, which they propose to represent an impairment in late endosomes/MVBs trafficking to lysosomes. These vesicles accumulate TDP43-GFP, but only upon treatment with the lysosome inhibitor, HCQ, suggesting that TDP43-GFP is being degraded (or the green fluorescence quenched) in this compartment. However, upon NEDD4 knockdown, TDP43-GFP does not accumulate in HCQ-treated Rab7 vesicles. Based on these results, the authors propose that NEDD4-mediated ubiquitination of TDP43 results in its ESCRT-dependent incorporation into intraluminal vesicles, resulting in its endolysosomal clearance in process that resemble endosomal micro-autophagy (eMI). Although the genetics support a role for ESCRTs and NEDD4 in cytoplasmic TDP43 clearance, the cell biological studies are low resolution in scope. The author's model requires additional experimental support that TDP43 is being incorporated into intraluminal vesicles (ILVs).

1) Although Rab7 localizes to late endosomes/MVBs, it is not uniquely present at this compartment in cells. Thus, additional evidence is needed to support that the large vesicles in TDP43 overexpressing cells are late endosomal in origin. Additional LE/MVB markers such as CD63 and LBPA should be analyzed in Figures 5 and 6.

Response: This is a good suggestion, similar to point 7 raised by reviewer 1. We additionally probed our Rab7 giant vesicular structures with an antibody against CD63 (another MVB marker). In new data (Supplemental Figure 4C), we show co-localization of CD63 and Rab7 in WT and TDP-43-GFP expressing cells. We have also quantified this data (Supplementary Table 2). This further supports the assertion that these compartments are MVB-like in nature, at least based on protein composition.

2) Transmission electron microscopy of the Rab7 positive vesicles in TDP43- or TDP35-overexpressing cells should be conducted to confirm that these organelles exhibit evidence of ILVs as proposed by the authors in their model. Furthermore, immuno-EM using GFP antibodies will help corroborate that TDP-43 is located at these ILVs. These are critical experiments to corroborate that the large vacuoles induced by TDP43 are MVBs that contain ILVs with TDP-43 as cargo.

Response: This is a good suggestion, also raised by reviewer 1 (point 8). Please see our detailed response there.

3) Similarly, for the studies of NEDD4 siRNA in Fig 5A, the effects of CHQ treatment on TDP-43 protein turnover following cycloheximide treatment should be analyzed via immunoblotting to confirm the role of endolysosomal acidification on the TDP43 and TDP35 clearance.

Response: This is a good idea, but, respectfully, we decided there were sufficient other datasets in our paper with CHQ that reasonably argued that NEDD4 promoted TDP-43 turnover in a manner dependent

on endolysosomal acidification. First, we have shown that CHQ causes TDP-43 accumulation in a Rab7+ve vesicles (Figure 8B). Second, our new data (Supplemental Figure 3) indicates that NEDD4 impacts interaction of TDP-43 with Vps4 (suggesting endolysosomal targeting). Third, our new data (Supplemental Figure 2) indicates that TDP-43 is modified with K63-linked polyubiquitin (also suggesting endolysosomal targeting).

4) Immunofluorescence analysis for K63-ubiquitin chains within these large vesicle should be performed in Fig 7B-C to potentially support the author's model.

Response: This is an interesting idea, however we would note that it is unclear as to whether ubiquitin is always removed from substrates prior to targeting within MVBs (e.g. Huebner et al, Pistikun, 2016 Mol Cell Proteomics, and refs within), so it is not clear that we would expect to see K63-Ub signal within these large vesicles in the first place. Nonetheless, this is a subject we hope to explore in future studies.

5) Does NEDD4 and VPS4 control TDP43 secretion into the conditioned media via EVs? In eMI, substrates are not only degraded but secreted outside of a cell via extracellular vesicles (EVs).

Response: This is a good point, and one that we hope to investigate in the future as part of a future manuscript. Respectfully, we think it is beyond the scope of this study which is focused on TDP-43 degradation.

6) The analysis of "cell viability" in Fig 3D and 4C is based on enumerating cell numbers. However, this does not distinguish between reduced cell proliferation versus increased cell death in these cultures. More precise measures of cell death, either measurement of apoptotic markers or dye exclusion in cells, should be analyzed to draw conclusions regarding cell viability or survival in these assays.

Response: We fully agree with this point; we have changed the term "viability" to "proliferation" where necessary (p10, results text; p11, results text; p25 methods; p35 Fig 4 legend; p46 Fig 5 legend; Fig 4 axes; Fig 5C axes).

Minor issues:

1) The cycloheximide chase immunoblots in Fig 3A are not particularly convincing for turnover for TDP-43 the controls. Longer chase periods (72h) should be considered.

Response: We agree; in fact this was an unintentional error in the figure preparation process, in which the 72hr data was excluded. We now include this, with extended quantitation, in Fig 4A and in our supplemental table 2.

2) The immunoblot in Fig 4B is not particularly convincing for a reduction in TDP43 following NEDD4 overexpression. A cycloheximide chase should be performed similar to the NEDD4 knockdown.

Response: Note for clarity, we believe the reviewer is referring to Figure 5B. Regardless, we did perform a CHX pulse chase with NEDD4 KD; this is shown in Figure 5A. Note that the efficiency of NEDD4 KD is not especially high (compare NEDD4 blot at any given timepoint with control siRNA timepoint, and see quantitation in Supplemental Table 2). Still, we see a statistically significant stabilization of TDP-43 half-life following NEDD4 KD.

Other changes (not requested by reviewers; made to increase clarity, accuracy, relevance etc)

We split the results section on NEDD4 into two distinct sections; the first focused on NEDD4 interactions with TDP-43, and its impacts on TDP-43 stability and toxicity. The second focuses on NEDD4 ubiquitination of TDP-43, including our new K63 and K48 datasets (New Supplementary Figure 2)

We have added clarifying detail to figure legends regarding which plasmids were utilized in our experiments, and added previously missing details of control plasmid vectors (plasmid numbers reference details present in Supplemental Table 3).

We improved consistency in terms used throughout the paper.

We have added new discussion material to bring our work “up to date” since our last submission (e.g. TDP-43 being a putative NEDD4 substrate based on a Ub proteomics study).

We have removed the term “giant MVB” in the abstract, intro (final paragraph) results text and discussion, and primarily highlighted their size and composition (Rab7 and CD63 positive). This in part reflects the fact we already interchangeably used the term giant “Rab7-positive vesicle” and giant MVB, which was potentially confusing, and because ILVs are not consistently observed in our data, suggesting that what we are observing, whilst compositionally resembling MVBs, may not morphologically or functionally fully resemble them.

July 22, 2025

Re: JCB manuscript #202212064R

J. Buchan
University of Arizona

Dear Dr. Buchan,

Thank you for submitting your revised manuscript entitled " Rsp5/NEDD4 and ESCRT regulate TDP-43 toxicity and turnover via an endolysosomal clearance mechanism" to Journal of Cell Biology. The manuscript has been re-assessed by two of the original reviewers, whose reports are appended below (unfortunately, reviewer #1 never responded to our requests for re-review).

As you will see, although reviewer #2 is now supportive of publication, reviewer #3 continues to feel that the main conclusions remain insufficiently supported. And these issues, combined with our own editorial concerns about the revision, mean that we are unable to consider the paper further.

As you may know, JCB policy is such that papers are considered through only one major revision cycle and, while it may be theoretically feasible to address these remaining concerns, it would require major revisions. Thus, we unfortunately cannot offer publication of the manuscript.

If you would like to transfer your reviewer comments to another journal for consideration elsewhere, please contact the journal office at cellbio@rockefeller.edu and we would be happy to arrange the transfer on your behalf.

While we cannot pursue this manuscript further, we encourage you to transfer your study to our not-for-profit open-access sister journal, Life Science Alliance (LSA). We shared your manuscript and the accompanying reviews with LSA Executive Editor, Tim Fessenden, who is interested in these findings. He is pleased to offer publication of this manuscript at LSA pending the following revisions: - Temper claims on the presence of TDP-43 in intraluminal vesicles as remarked by Reviewer 3 in point 1. - Temper claims on the requirement of lysosomal function to clear TDP-43 in a NEDD4-dependent manner in Figure 5, as remarked by Reviewer 3 in point 2. We encourage you to use the link below to transfer your manuscript to LSA. You do not need to revise the manuscript before transferring it to LSA. Once you transfer, Dr. Fessenden will email you an invitation to revise and resubmit, listing the same revision requests as mentioned above. Please feel free to reach out at t.fessenden@life-science-alliance.org if you have any questions about the LSA journal, the transfer process, or the revisions requested.

Link Not Available

In any case, we truly appreciate the effort that has gone into the revisions and sincerely regret that the outcome is not more positive.

Sincerely,

Harald Stenmark, PhD
Monitoring Editor
Journal of Cell Biology

Tim Spencer, PhD
Executive Editor
Journal of Cell Biology

Reviewer #2 (Comments to the Authors (Required)):

The authors have been responsive to my review and addressed the majority of my questions/suggestions.

Reviewer #3 (Comments to the Authors (Required)):

I have reviewed the manuscript and their response to my previous critiques. The authors have attempted to answer several questions but certain key concerns persist.

1) An important concern raised was whether the large Rab7 vesicles in TDP43 overexpressing cells are late endosomal in origin (points 1 and 2 in my previous critique). As the authors indicate, attempts to perform EM to confirm the presence of ILVs within Rab7 positive vesicles has not proven successful. However, their new data with CD63 staining in Supplemental Figure 4C, CD63 is located only in the limiting membrane but not within the ILVs in the lumen of these structures. Based on the authors' proposed model, if these vesicles incorporate TDP43 via intraluminal budding, CD63 should be present on ILVs within these structures. LBPA immunostaining was requested as an alternative method to detect ILVs but was not performed. Evidence for TDP43 incorporation within ILVs is important to support the key conclusion in the paper that the large vesicles induced by TDP43 accumulation are MVB-like. Thus far, no convincing evidence has been provided.

2) In response to point 3, the authors argue against the need for performing cycloheximide chase assay using a lysosomal inhibitors in Fig 5A arguing that the HCQ treatment studies in Fig. 8 sufficiently demonstrate that TDP43 accumulates in Rab7 vesicles in response to HCQ and that this phenotype is NEDD4 dependent. Although the images do support this conclusion, the amount of TDP43-GFP present within Rab 7 structures is quite modest compared to the overall pool because of its expected location in the nucleus; the statistical analysis in Fig 8C quantifies the percentage of cells with intraluminal TDP43-GFP but this metric does not provide insight into the overall flux of TDP43-GFP in the endolysosomal compartment. Moreover, in the whole cell lysates provided as controls in Supplemental Fig 3 indicates that endogenous TDP43 levels is not impacted by either HCQ treatment or NEDD4 knockdown, although this may pathway may uniquely manifest in the context of TDP43 accumulation or overexpression. Nevertheless, biochemical analysis of the overall pool of TDP43 to scrutinize the functional role of endolysosomal function in the clearance of TDP-43 is missing. Hence, in this reviewer's view, conducting the requested TDP43 cycloheximide chase assay in Fig 5A in the presence versus absence of a lysosomal inhibitor remains important to rigorously test the proposed model that clearance of excess cytoplasmic TDP43 is mediated by the endolysosome.

Summary of manuscript changes:

1. New data panels in Supplementary Figure 4 (B-E) with accompanying results text showing that giant Rab7-positive vesicles can contain luminal CD63 signal (likely ILVs), especially if endolysosomal degradation is impeded. TDP-43 is also detected in association with such intra-luminal CD63 vesicles/puncta. This addresses Reviewer 3's concerns about the nature of the giant Rab7 vesicles and TDP-43's potential entry mechanism.
2. Associated data added to Supplementary Table 2 (data quantification)
3. Moved data to Figure 7 (from previous version of Supplemental Figure 4) to accommodate the new data above.
4. Significant discussion section added on the importance of investigating how our degradation pathway may influence TDP-43 secretion.
5. Future directions concerning the study of the endolysosomal compartment TDP-43 enters, through greater compositional analysis and high resolution (e.g., electron microscopy) based methods, was also added to the discussion.
6. Methods section detail concerning our mammalian cell IF data analysis strategies has been added (this was erroneously missing before).
7. Various clarifying text edits throughout the manuscript to better describe our working model, as well as our prior data, and that of others, that supports the existence of TDP-43 endolysosomal degradation.
8. Updated bibliography, figure legends and acknowledgements.

Reviewer comments and our responses (in blue italics)

Reviewer #3: - First revision round comments

1) Although Rab7 localizes to late endosomes/MVBs, it is not uniquely present at this compartment in cells. Thus, additional evidence is needed to support that the large vesicles in TDP43 overexpressing cells are late endosomal in origin. Additional LE/MVB markers such as CD63 and LBPA should be analyzed in Figures 5 and 6.

Appeal Response: As noted in our last submission/response to reviewers document, we probed our Rab7 giant vesicular structures with an antibody against CD63 (another MVB marker) and showed (and quantified) co-localization of CD63 and Rab7 in WT and TDP-43-GFP expressing cells. We attempted immunostaining with an LBPA Ab, but control experiments indicated unreliable antibody performance.

2) Transmission electron microscopy of the Rab7 positive vesicles in TDP43- or TDP35-overexpressing cells should be conducted to confirm that these organelles exhibit evidence of ILVs as proposed by the authors in their model. Furthermore, immuno-EM using GFP antibodies will help corroborate that

TDP-43 is located at these ILVs. These are critical experiments to corroborate that the large vacuoles induced by TDP43 are MVBs that contain ILVs with TDP-43 as cargo.

Appeal Response: As noted in our last submission/response to reviewers document, this was a good suggestion that we attempted; we were unsuccessful due to a lack of suitable core facility resources and skilled staff.

We have added text at the end of the 2nd paragraph in our discussion acknowledging that greater compositional and high-resolution microscopy methods such as electron microscopy are needed to fully understand the nature of the endolysosomal compartment TDP-43 enters into.

3) Similarly, for the studies of NEDD4 siRNA in Fig 5A, the effects of CHQ treatment on TDP-43 protein turnover following cycloheximide treatment should be analyzed via immunoblotting to confirm the role of endolysosomal acidification on the TDP43 and TDP35 clearance.

Appeal Response: In our last submission, we previously outlined data in our paper that strongly suggests NEDD4-mediated TDP-43 endolysosomal clearance is occurring (in addition to the clear involvement of ESCRT proteins in turnover, whose role in endolysosomal entry is generally not in dispute).

We have also added new text that we previously showed association of TDP-43 with early and late endosomes based on fractionation studies to our introduction (5th paragraph, includes references to other papers, besides ours, supporting endolysosomal degradation of TDP-43). Despite this, our discussion also acknowledges that NEDD4 may promote TDP-43 degradation by other means.

4) Immunofluorescence analysis for K63-ubiquitin chains within these large vesicle should be performed in Fig 7B-C to potentially support the author's model.

Appeal response: In our final discussion paragraph, we have added new text acknowledging the importance of determining how TDP-43 modification status ultimately influences TDP-43 degradation (or secretion).

We did confirm in the first revision that endogenous TDP-43 is subject to K63 ubiquitination by NEDD4. Otherwise, our prior reviewer response notes that Ub may not be expected to be observed in MVB/giant Rab7 vesicles anyway, meaning the IF datasets the reviewer requests would not be straightforward to interpret.

5) Does NEDD4 and VPS4 control TDP43 secretion into the conditioned media via EVs? In eMI, substrates are not only degraded but secreted outside of a cell via extracellular vesicles (EVs).

Appeal response: This is an important area of future investigation. We have added extensive new text (>250 words) and 10 new references highlighting the fact that MVBs can secrete their contents via exocytosis, that neurodegenerative disease proteins can be subject to this, and that various factors have been identified as potential regulators of the process. We also discussed a distinct secretion mechanism called MAPS (misfolding associated protein secretion pathway) which shares some machinery with endosomal microautophagy and has also been reported to impact TDP-43 secretion. We believe this is a valuable addition to the discussion and indeed is an area of future research for us.

6) The analysis of "cell viability" in Fig 3D and 4C is based on enumerating cell numbers. However, this does not distinguish between reduced cell proliferation versus increased cell death in these cultures. More precise measures of cell death, either measurement of apoptotic markers or dye exclusion in cells, should be analyzed to draw conclusions regarding cell viability or survival in these assays.

Appeal response: Nothing to add from our previous response. We agreed fully with the reviewer and made the changes they asked for.

Minor issues:

1) The cycloheximide chase immunoblots in Fig 3A are not particularly convincing for turnover for TDP-43 the controls. Longer chase periods (72h) should be considered.

Appeal response: Nothing to add from our previous response. We agreed fully with the reviewer and made the changes they asked for.

2) The immunoblot in Fig 4B is not particularly convincing for a reduction in TDP43 following NEDD4 overexpression. A cycloheximide chase should be performed similar to the NEDD4 knockdown.

Appeal response: Nothing to add from our previous response.

Reviewer #3: - Second revision round comments

I have reviewed the manuscript and their response to my previous critiques. The authors have attempted to answer several questions but certain key concerns persist.

1) An important concern raised was whether the large Rab7 vesicles in TDP43 overexpressing cells are late endosomal in origin (points 1 and 2 in my previous critique). As the authors indicate, attempts to perform EM to confirm the presence of ILVs within Rab7 positive vesicles has not proven successful. However, their new data with CD63 staining in Supplemental Figure 4C, CD63 is located only in the limiting membrane but not within the ILVs in the lumen of these structures. Based on the authors' proposed model, if these vesicles incorporate TDP43 via intraluminal budding, CD63 should be present on ILVs within these structures. LBPA immunostaining was requested as an alternative method to detect ILVs but was not performed. Evidence for TDP43 incorporation within ILVs of is important to support the key conclusion in the paper that the large vesicles induced by TDP43 accumulation are MVB-like. Thus far, no convincing evidence has been provided.

Appeal response: New data description

The reviewer suggests that CD63-positive ILV structures should be evident in our giant Rab7-positive vesicles if they are of endolysosomal origin (i.e. MVB-like). In our previous submission we had focused on the CD63-Rab7 co-localization data that now forms panel A of Supplemental 4, as this was the most common morphological phenotype in TDP-43-GFP expressing cells under normal conditions, and we thought it answered the reviewers original point 1, which had made no mention of quantifying ILVs. However, we also occasionally observed other phenotypes including CD63 puncta (presumed ILVs) within

the Rab7 vesicle lumen (most commonly near the Rab7 membrane – see Supp Fig 4B, top row). Even more rarely, we observed TDP-43-GFP puncta, adjacent or partially enveloped by CD63 signal, both of which lay within Rab7 vesicular lumen (Supp Fig 4B, 2nd row). This suggested two important points:

1. That Rab7-positive vesicles harbored a low level of CD63-positive ILVs, and;
2. TDP-43 detection within Rab7 vesicles, co-localized with CD63, suggests an internalization mechanism reminiscent of ESCRT-driven events that occur at MVBs.

We suspected that the relatively low levels of these phenotypes reflected degradation within the giant Rab7-positive vesicle and thus repeated our analysis with CHQ present to inhibit this process. Importantly, we saw:

1. That the percentage of CD63 signal (presumed ILVs) within the lumen of Rab7-positive vesicles increased significantly (e.g., Supp Fig 4B, row 4, green arrows; quantified in Supp Fig 4C; nearly 4-fold increase).
2. The amount of TDP-43 puncta colocalized or enveloped by CD63, detected within Rab7 vesicular lumens, also increased strongly (e.g. Supp Fig 4B, row 4; quantified in Supp Fig 4E).

These observations are fully consistent with the idea that Rab7-positive vesicles are at least partially MVB-like and that TDP-43 enters via an ESCRT-driven mechanism typical of MVB function, and that Rab7-positive vesicles are proteolytically active (consistent with our data in Figure 8 also).

We have provided reasons in a prior e-mail to the editors as to why this data was not included in our first revision, but are happy to include it now and discuss it fully.

We have summarized these observations, and our interpretations, in the section **“TDP-43 accumulation causes appearance of enlarged Rab7/CD63 positive vesicular organelles”** – 4th paragraph. We also made minor clarity adjustments in earlier paragraphs (e.g., to make clearer why we hypothesized that TDP-43 might be targeted to MVBs via an ESCRT driven process; also references to our earlier work where endolysosomal TDP-43 degradation was previously described). We also add clarifying statements in the 5th paragraph regarding our conclusions as to the nature of the giant Rab7-positive vesicles.

In the section **“TDP-43 accumulates within giant Rab7-positive vesicles in a NEDD4-dependent manner”**, we add a clarifying statement in the first paragraph stating that we do not think TDP-43-GFP is primarily degraded in giant Rab-7 vesicles, for the simple reason they are not present in most cells. In the second paragraph, we describe our data showing TDP-43 localization with CD63, within Rab7-vesicles, and suggest this supports TDP-43 internalization via an ESCRT-dependent mechanism similar to what one would expect at canonical MVBs.

2) In response to point 3, the authors argue against the need for performing cycloheximide chase assay using a lysosomal inhibitors in Fig 5A arguing that the HCQ treatment studies in Fig. 8 sufficiently demonstrate that TDP43 accumulates in Rab7 vesicles in response to HCQ and that this phenotype is NEDD4 dependent. Although the images do support this conclusion, the amount of TDP43-GFP present within Rab 7 structures is quite modest compared to the overall pool because of its expected location in

the nucleus; the statistical analysis in Fig 8C quantifies the percentage of cells with intraluminal TDP43-GFP but this metric does not provide insight into the overall flux of TDP43-GFP in the endolysosomal compartment. Moreover, in the whole cell lysates provided as controls in Supplemental Fig 3 indicates that endogenous TDP43 levels is not impacted by either HCQ treatment or NEDD4 knockdown, although this may pathway may uniquely manifest in the context of TDP43 accumulation or overexpression. Nevertheless, biochemical analysis of the overall pool of TDP43 to scrutinize the functional role of endolysosomal function in the clearance of TDP-43 is missing. Hence, in this reviewer's view, conducting the requested TDP43 cycloheximide chase assay in Fig 5A in the presence versus absence of a lysosomal inhibitor remains important to rigorously test the proposed model that clearance of excess cytoplasmic TDP43 is mediated by the endolysosome.

Appeal response (largely the same as in e-mail to Drs Spencer and Stenmark, July 24th): Putting aside the fact we did not believe any new experimental data was required for this point, some concerns here are new from the previous critique. For example, measuring flux of TDP-43 into an organelle is no mean technical feat. Regarding the role of endolysosomal function in the clearance of TDP-43, our 2017 Nature Communications paper (and work of other labs since e.g. Ichida, Braun, Eggan) has established the existence of this mechanism, as we noted in our introduction (paragraph 5).

New: we would add that we demonstrate in this paper that Rsp5-driven degradation of TDP-43 is ESCRT dependent, and that NEDD4 activity promotes TDP-43 interactions with VPS4 (ESCRT factor). A large body of existing evidence in the literature also suggests that ESCRT-driven degradation events are dependent on the lysosome.

October 3, 2025

RE: JCB Manuscript #202212064RR-A

J. Buchan
University of Arizona

Dear Dr. Buchan:

Thank you for submitting your revised manuscript entitled "Rsp5/NEDD4 and ESCRT regulate TDP-43 toxicity and turnover via an endolysosomal clearance mechanism". We have now gone through the revised paper and your response to the remaining reviewer comments and we would be happy to publish your paper in JCB pending final revisions necessary to meet our formatting guidelines (see details below).

A. MANUSCRIPT ORGANIZATION AND FORMATTING:

1) Text limits: Character count for Articles is typically < 40,000, not including spaces. Count includes the abstract, introduction, results, discussion, and acknowledgments. Count does not include title page, materials and methods, figure legends, references, tables, or supplemental legends. Your paper is just under this limit - if you find that you need to add a bit more text and exceed the 40,000 characters, that is fine in this case, but please try to be as concise as possible.

2) Figure formatting: Scale bars must be present on all microscopy images, including inset magnifications. Molecular weight or nucleic acid size markers must be included on all gel electrophoresis. Therefore, you must add molecular weight markers to the gels/blots in figures 1D, 2A, 2C, 3C, 3F, 4A, 5A, 5B, 5E, 5F, Supplemental figures 2A-B, and 3.

3) Statistical analysis: Error bars on graphic representations of numerical data must be clearly described in the figure legend. The number of independent data points (n) represented in a graph must be indicated in the legend. Statistical methods should be explained in full in the materials and methods in a separate section which describes all statistical methodology employed. For figures presenting pooled data the statistical measure should be defined in the figure legends. Please also be sure to indicate the statistical tests used in each of your experiments (both in the figure legend itself and in a separate methods section) as well as the parameters of the test (for example, if you ran a t-test, please indicate if it was one- or two-sided, etc.).

****Also, since you used parametric tests in your study (e.g. t-tests, ANOVA, etc.), you should have first determined whether the data was normally distributed before selecting that test. In the stats section of the methods, please indicate how you tested for normality. If you did not test for normality, you must state something to the effect that "Data distribution was assumed to be normal but this was not formally tested."****

4) Materials and methods: Should be comprehensive and not simply reference a previous publication for details on how an experiment was performed. Please provide full descriptions (at least in brief) in the text for readers who may not have access to referenced manuscripts. The text should not refer to methods "...as previously described."

5) Please be sure to provide the sequences for all of your primers/oligos and RNAi constructs in the materials and methods. You must also indicate in the methods the source, species, and catalog numbers (where appropriate) for all of your antibodies.

6) Microscope image acquisition: The following information must be provided about the acquisition and processing of images:

- a. Make and model of microscope
- b. Type, magnification, and numerical aperture of the objective lenses
- c. Temperature
- d. imaging medium
- e. Fluorochromes
- f. Camera make and model
- g. Acquisition software
- h. Any software used for image processing subsequent to data acquisition. Please include details and types of operations involved (e.g., type of deconvolution, 3D reconstitutions, surface or volume rendering, gamma adjustments, etc.).

7) References: There is no limit to the number of references cited in a manuscript. References should be cited parenthetically in the text by author and year of publication. Abbreviate the names of journals according to PubMed.

8) Supplemental materials: There are typically strict limits on the allowable amount of supplemental data. Articles may have up to 5 supplemental figures. Your paper current has 7 such figures but please note that the full length blots should not be provided as supplementary figures but, instead, as "Source Data" (see the instructions regarding source data further down in this list). Therefore, you can move these final two supplemental figures to the Source Data (and also remove the associated figure legends).

Please also note that tables, like figures, should be provided as individual, editable files. A summary of all supplemental material (that is, in addition to the supplementary figure legends) should appear at the end of the Materials and methods section. Please see any recent JCB paper for an example of this.

9) eTOC summary: A ~40-50 word summary that describes the context and significance of the findings for a general readership should be included on the title page. The statement should be written in the present tense and refer to the work in the third person.

****We realize that you have already provided a summary but please edit to "First author name(s) et al..." to match our preferred style (rather than using "we" or "our").****

10) Conflict of interest statement: JCB requires inclusion of a statement in the acknowledgements regarding competing financial interests. If no competing financial interests exist, please include the following statement: "The authors declare no competing financial interests." If competing interests are declared, please follow your statement of these competing interests with the following statement: "The authors declare no further competing financial interests."

11) A separate author contribution section is required following the Acknowledgments in all research manuscripts. All authors should be mentioned and designated by their first and middle initials and full surnames. We encourage use of the CRediT nomenclature (<https://casrai.org/credit/>).

12) ORCID IDs: ORCID IDs are unique identifiers allowing researchers to create a record of their various scholarly contributions in a single place. Please note that ORCID IDs are now ***required*** for all authors. At resubmission of your final files, please be sure to provide your ORCID ID and those of all co-authors.

13) Please note that JCB now requires authors to submit Source Data used to generate figures containing gels and Western blots with all revised manuscripts. This Source Data consists of fully uncropped and unprocessed images for each gel/blot displayed in the main and supplemental figures. For assays performed using capillary electrophoresis and/or immunoassay-based detection, authors should instead provide the electropherogram graph(s) for each experiment, plotting fluorescence/chemiluminescence intensity vs. molecular weight/size. Since your paper includes cropped gel and/or blot images, please be sure to provide one Source Data file for each figure gels, blots, and/or capillary electrophoresis assays along with your revised manuscript files. File names for Source Data figures should be alphanumeric without any spaces or special characters (i.e., SourceDataF#, where F# refers to the associated main figure number or SourceDataFS# for those associated with Supplementary figures). For traditional gels and blots, the lanes of the gels/blots should be labeled as they are in the associated figure, the place where cropping was applied should be marked (with a box), and molecular weight/size standards should be labeled wherever possible. For capillary electrophoresis assays, each trace in the graph should be color-coded and labeled to indicate which protein, gene, or sample is being measured (please try to avoid red/green combinations to accommodate our color-blind readers).

14) Journal of Cell Biology now requires a data availability statement for all research article submissions. These statements will be published in the article directly above the Acknowledgments. The statement should address all data underlying the research presented in the manuscript. Please visit the JCB instructions for authors for guidelines and examples of statements at (<https://rupress.org/jcb/pages/editorial-policies#data-availability-statement>).

B. FINAL FILES:

Thank you for your attention to these final processing requirements. Please revise and format the manuscript and upload materials within 7-14 days. If you need an extension for whatever reason, please let us know and we can work with you to determine a suitable revision period.

Thank you for this interesting contribution, we look forward to publishing your paper in Journal of Cell Biology.

Sincerely,

Harald Stenmark, PhD
Monitoring Editor
Journal of Cell Biology

Tim Spencer, PhD
Executive Editor
Journal of Cell Biology